# APRES: AN AGENTIC PAPER REVISION AND EVALUATION SYSTEM

## ABSTRACT

Scientific discoveries must be communicated clearly to realize their full potential. Without effective communication, even the most groundbreaking findings risk being overlooked or misunderstood. The primary way scientists communicate their work and receive feedback from the community is through peer review. However, the current system often provides inconsistent feedback between reviewers, ultimately hindering the improvement of a manuscript and limiting its potential impact. In this paper, we introduce a novel method APRES powered by Large Language Models (LLMs) to update a scientific paper's text based on an evaluation rubric. Our automated method discovers a rubric that is highly predictive of future citation counts, and integrate it with APRES in an automated system that revises papers to enhance their quality and impact. We demonstrate the success of APRES, which improves future citation prediction by 19.6% in mean averaged error over the next best baseline, and show that our paper revision process yields papers that are preferred over the originals by human expert evaluators 79% of the time. Our findings provide strong empirical support for using LLMs as a tool to help authors "stress-test" their manuscripts before submission. Ultimately, our work seeks to augment, not replace, the essential role of human expert reviewers, for it should be humans who discern which discoveries truly matter, guiding science toward advancing knowledge and enriching lives.

## 1 INTRODUCTION

Peer review stands as the cornerstone of scholarly communication, a critical process for validating the quality, novelty, and correctness of scientific research. However, this essential system is currently under unprecedented strain (Kim et al., 2025). Top-tier conferences now receive tens of thousands of submissions annually, a volume that vastly outpaces the growth of the available qualified reviewer pool. This has led to significant challenges, including reviewer fatigue, lengthy review cycles, and, most critically, concerns about the consistency of evaluation. These challenges lead to a difficult scenario for authors who wish to communicate their scientific findings and receive feedback on how to further improve their work (Chen et al., 2025a). Large Language Models (LLMs) offer a powerful new avenue for providing authors, especially non-native speakers, with scalable feedback. However, their direct application for paper review and revision carries significant risks, such as the potential to inadvertently modify scientific claims or deviate from accepted academic styles (Ye et al., 2024; Lin et al., 2025). We address this critical gap by developing a constrained technique for LLM scientific paper reviewing and revising. We develop a constrained LLM-based approach for scientific paper reviewing and revising that focuses on strengthening a paper's presentation and readability.

We introduce a novel agentic framework, called APRES, that leverages LLMs not only to mimic human review, but to first discover what evaluation criteria is predictive of a paper's future impact. APRES then uses that knowledge to guide an automated revision process to help authors improve their work. Our primary contribution is a two-stage method. First, we employ an agentic search scaffold (Jiang et al., 2025; Zhao et al., 2025a) to discover an optimal review rubric by training a negative binomial regression model (Lawless, 1987) to directly predict citation counts from discovered rubric scores, moving beyond fixed, human-defined criteria. Second, we use this discovered rubric as an objective function in a closed-loop approach where a 'Rewriter' selectively revises a paper's text to maximize its predicted impact score.

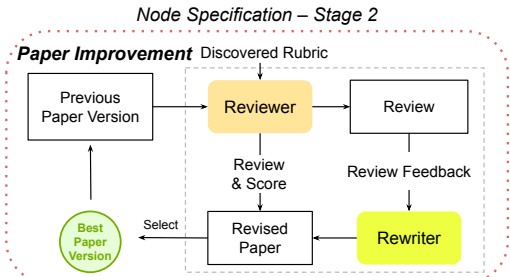

Figure 1: The general search framework and the node specification for *Stage 1 - Rubric Search* of APRES. APRES utilizes the same agentic search scaffold (left) for two distinct tasks. The right details the inner-loop logic of the first stage for rubric discovery. Each node in stage 1 takes the previous identified best rubric and the human papers as input, and then proposes a new rubric for scoring. The scores of the new rubric are used to train a negative binomial regression model to predict the number of citations. If the newly proposed rubric obtains a better score than the previous best rubric, it is selected as the new best one.

To validate APRES, we report results of our rubric search and automated revision approaches, demonstrating that they can effectively predict impact and guide meaningful improvements. In the appendix, we further demonstrate that the LLM evaluation pipeline is reliable by replicating the design of the NeurIPS consistency studies (Cortes & Lawrence, 2021; Beygelzimer et al., 2023). Through this work, we demonstrate that LLMs, guided by agentic methods, can pave the way for a more reliable and data-driven processes to help authors improve their scientific communication.

Our findings contribute to a crucial and timely discussion as the scientific community begins to experiment with integrating AI into the review process. At the time of writing, conferences like AAAI (AAAI 2026) have begun piloting programs to provide authors with supplementary AI-generated reviews, while ICLR (Thakkar et al., 2025) has explored using LLMs to provide real-time feedback to human reviewers to help improve the constructiveness of their comments. Our work provides strong empirical evidence that LLMs can serve as a reliable tool to reduce the inherent randomness of peer review and can even uncover novel signals of a paper's future impact that elude traditional review criteria. For predicting future impact of a paper, our discovered rubric yields a 19.6% improvement in the Mean Averaged Error (MAE) score over the next best baseline. For paper improvement, APRES revised papers are preferred over the human written original

Figure 2: The node specification for *Stage 2 - Paper Improvement*. Each node in stage 2 takes the previous best paper revision and the discovered rubric as input, and then runs the paper with the rubric through the reviewer model to generate feedback to the Rewriter to revise the paper again. The newly revised paper is scored by the same reviewer again, and if it scores better than the previous one it becomes the new best paper.

paper 79% of the time. However, this does not mean we advocate for fully autonomous AI review systems. Significant challenges remain, including the potential for manipulation through adversarial methods like prompt injection, where hidden instructions in a manuscript could skew an LLM's evaluation. Instead, we envision our work as paving the way for sophisticated human-in-the-loop systems, where AI agents provide a consistent, data-driven baseline that empowers authors to improve their paper's quality and readability, while assisting reviewers in making more informed and equitable decisions.

## 2 RELATED WORK

**LLMs for review generation and author assistance.** Early attempts to automate reviewing used information extraction and template-based methods to produce comments (Wang et al., 2020), or

studied the feasibility of automatic review generation more broadly (Yuan et al., 2022). With the advent of instruction-tuned LLMs, research has shifted to more structured review generation pipelines. For example, Gao et al. (2024) improves coverage of the range of the opinions that human reviewers produce by explicitly prompting across multiple aspects, while multi-agent and tree-of-thought frameworks (D'Arcy et al., 2024; Chang et al., 2025) aim to emulate the deliberative process of human committees. Zeng et al. (2025) used reinforcement learning to learn a reviewer to generate useful reviews. Large-scale empirical studies indicate that LLM feedback substantially overlaps with human reviews, with agreement levels comparable to inter-human reviewer consistency (Liang et al., 2023), although controlled experiments suggest its utility remains modest in real reviewing workflows (Robertson, 2023). More practically, LLM-based checklist assistants have been deployed to help authors align with venue guidelines, and were found to lead to 70% of all surveyed authors to revise their submitted paper in the NeurIPS 2024 pilot study (Goldberg et al., 2024). In ICLR 2025, it was shown that LLM provided review feedback helped reviewers provide more specific and actionable reviews (Thakkar et al., 2025). Choi et al. (2025) also presents a position that LLMs should not be used simply for text generation in the review process, but should be utilized for more meaningful tasks such as checking the relevance of citations, ethic review flagging, or verifying experimental results. We compare our approach to existing methods in Tab. A1.

**AI-modified reviews and detection.** The increasing integration of LLMs into reviewing raises new concerns about provenance and authenticity. Recent analyses estimate that a non-trivial fraction of reviews submitted to AI conferences show signs of LLM modification (Liang et al., 2024). To address this, tailored detectors have been benchmarked on peer-review corpora (Yu et al., 2025), and classifiers trained to distinguish LLM-written reviews from human ones have been proposed (Rao et al., 2025). ReviewScore (Ryu et al., 2025) focuses on detecting misinformed reviews using LLMs, which is an orthogonal task to our work. However, detection remains unreliable at the individual level, suggesting that transparency and disclosure policies are necessary if AI assistance in reviewing becomes widespread.

**Reliability of peer review.** Our work is also connected to long-standing concerns over the reliability of peer review. Landmark experiments have demonstrated the arbitrariness of some decisions, with two independent committees disagreeing on roughly a quarter of papers in both NeurIPS 2014 and 2021 (Cortes & Lawrence, 2021; Beygelzimer et al., 2023). Subsequent studies have examined alignment between review text and numerical scores, finding that written critiques generally correlate with ratings and confidence, with some systematic biases (Wu et al., 2024). Such findings underscore the importance of designing reviewing pipelines that are consistent and objective.

**Impact prediction and scholarly recommendation.** Our work relates to bibliometric models that aim to predict a paper's future influence. Traditional approaches rely on metadata and citation graphs (Geng et al., 2022), whereas more recent methods incorporate document embeddings (Cohan et al., 2020a). With the rise of LLMs, purely text-based impact forecasting has become feasible, e.g., Zhao et al. (2025b) train LLMs on new articles to predict normalized citations, while Hao et al. (2024) combine retrieval with generative ranking to identify core citations, achieving notable gains across diverse fields. These works demonstrate that language-based features can capture impact signals complementary to citation networks, though it is important to emphasize that citation counts remain an imperfect proxy for scientific quality (Chen et al., 2025b).

Certain writing styles that helps readers quickly grasp the core idea of a papers also tend to give papers advantages of higher impact (Letchford et al., 2015; 2016). Ryba et al. (2021) showed that papers written in a more accessible style resulted in higher readability, understanding, and confidence. Ante (2022) studied this question from another perspective and conclude that lexical complexity in abstracts correlates positively with citations, especially in mature fields. In our work, we propose an agentic system that can automatically revise a paper's text by aiming to improve its predicted impact. This revision is not designed to only optimize the surface level simplicity, but to jointly emphasize a discovered rubric that consider the paper in whole for clarity of contribution, logical structure, and field relevance.

**Automated text revision for scientific writing.** LLMs have pushed automated text revision beyond simple grammatical error correction (Bryant et al., 2023) to more sophisticated systems that improve the clarity and quality of scientific papers (Jourdan et al., 2023). Early work explored revision mechanisms, which uses pretrained transformers for abstractive summarization with a remove/mask token strategy to revise generated output (Zhang et al., 2019). More recent approaches have lever-

aged multi-agent LLMs to provide constructive feedback for paper revisions (Chamoun et al., 2024) and have proposed human-in-the-loop pipelines to guide the revision process (Du et al., 2022).

**Peer-review corpora and resources.** A number of datasets have been released to study peer review at scale, e.g., (Kang et al., 2018; Lin et al., 2023; Zhou et al., 2024; Singh et al., 2021). While these datasets provide valuable insights into reviewer behavior and text characteristics, they are limited to past venues and older topic distributions. Instead, we curate a fresh evaluation set from publicly available papers and reviews from ICLR 2024 & 2025 and NeurIPS 2023 & 2024, ensuring that our benchmarks reflect the most current writing styles and review practices while minimizing potential data leakage from legacy datasets.

## 3 METHOD

The first contribution of this work is to develop a new LLM agentic framework that can provide informative feedback to authors to help them improve the quality and readability of their papers' text. This then supports the ultimate goal of our framework, which is to revise the original paper to make it more understandable with higher quality and clarity. To achieve this, a measure for the impact of the paper needs to be defined so that one can design a framework that can optimize the paper to maximize the impact. In this section, we detail our approach to design such a framework. An overview of APRES can be found in Fig. 1.

### 3.1 PREDICTING FUTURE IMPACT

Beyond quality control, a primary goal of peer review is to identify and champion research that will have a significant future impact. Traditionally, markers of high impact at conferences include prestigious oral or spotlight presentation slots. While these decisions serve as an important initial filter, they are not a perfect or stable proxy for a paper's long-term influence within the scientific community. A more direct, albeit lagging, measure of impact is a paper's citation count. We acknowledge that raw citation counts are an imperfect metric, as citation patterns can vary significantly across different subfields and research communities, and citation counts themselves can be artificially inflated or gamed. However, they remain one of the most widely used and available proxies for scientific influence (Dougherty & Horne, 2022; Lim et al., 2025). Therefore, a truly effective review system should not only align with conference decisions but also exhibit a predictive signal for future citations. This section details our approach for designing an agentic system that can extract such a signal for measuring impact from the paper content.

**Predicting Citations via Agentic Search.** We take the approach of a rubric search, where our agent is designed to search for rubric which when used to evaluate a paper yields scores that best predicts the citation count the paper receives twelve months after it becomes public. Citation data consists of non-negative integers and is often overdispersed, thus we use a negative binomial regression model to accurately capture the distribution of citations (Lawless, 1987).

The main challenge is determining the right features to feed into this regression model. Rather than relying on predefined rubric from conference guidelines, we designed an iterative search process to discover an informative set of rubric items for citation prediction. This process proceeds as follows:

1. **Propose:** An LLM agent, acting as a 'Rubric Proposer', proposes or refines a set of $k$ review rubric items, $\mathcal{R} = \{r_1, r_2, \ldots, r_k\}$, where each rubric item defines a specific aspect of a paper to evaluate (e.g., "originality", "clarity of presentation", "technical soundness", etc.).

2. **Score:** A second LLM agent, the 'Reviewer', scores every paper $p \in \mathcal{P}$ on each of the $k$ rubric items, producing a feature vector $\mathbf{x}_p = [s_{p,1}, s_{p,2}, \ldots, s_{p,k}]$, where each $s_{p,k}$ is a continuous score for each rubric item.

3. **Evaluate:** A negative binomial regression model is trained to predict the true citation count $y_p$ from the feature vector $\mathbf{x}_p$. The performance of this model is evaluated using the MAE metric.

4. **Select & Refine:** The performance score is logged, and the search process selects the best performing rubric so far and feeds it back to step one, which then modifies the rubric $\mathcal{R}$ in an attempt to improve the reviewer model's predictive accuracy in the next iteration.

This creates a closed loop that actively searches for the evaluation criteria most aligned with future scientific impact. We leverage the flexible search scaffold implementation from Zhao et al. (2025a) which extends upon AIDE search from Jiang et al. (2025). To be specific, this search scaffold

defines the **Select & Refine** step with four parameters, initial branch factor $N_0$, branch factor $N$, debug probability $p_{debug}$, and max debug depth $D_{max}$. The search in this scaffold starts with $N_0$ initial modifications to the initial rubric. At each subsequent step, a new set of rubric branches from either a randomly chosen buggy implementation (with probability $p_{debug}$) or the highest-performing rubric. Each set of rubric items would branch at most $N$ times. Debugging is capped at $D_{max}$ per node to avoid redundancy. In our rubric search, each of the 'Score' step is done by running a python script with the rubric defined by the agent, thus there maybe bugs in the code hence the need for the debug probability design. We name this search as MultiAIDE-Rubric.

## 3.2 USING LLM REVIEWS TO IMPROVE A PAPER'S CLARITY

Selecting papers that have higher chance of future impact is not the only goal for peer reviewing, it also indirectly serves the purpose of training authors to be able to present their work in a more easily understandable manner. A truly helpful review should provide constructive feedback that enables authors to meaningfully improve their work. To move beyond static evaluation and assess this practical utility, we propose a new approach to generate LLM feedback that can be operationalized by another LLM agent to verifiably enhance a paper's quality. This creates a fully automated, closed-loop system for critique and revision, allowing us to quantitatively measure the helpfulness of the initial review, as well as the ability of LLM systems to revise scientific papers.

To test this, we introduce an agent-based scaffold for iterative paper revision. The key idea is that although the actual impact of a paper is not known at the time of writing the paper, we can still use a surrogate function to approximate the paper impact and optimize with respect to that surrogate function. In this work, we use the rubric discovered by the previously introduced rubric search system as the surrogate function. The process leverages the final citation-predicting rubric, $\mathcal{R}^*$, discovered in our experiments with the procedure defined in Section 3.1, using it as the objective function to guide the revision process. The revision phase takes the form of a multi-step loop:

1. **Initial Scoring:** We begin with a paper, $p_{ori}$. An LLM 'Reviewer' agent generates a quantitative review with constructive feedback for the paper using the discovered rubric $\mathcal{R}^*$. We average the scores across all individual rubric items considered to obtain an overall score $S_{ori}$.

2. **Revision:** An LLM 'Rewriter' is tasked with improving the paper by proposing modifications to the current version of the paper based on the initial feedback from the LLM 'Reviewer'.

3. **Re-evaluation:** After modification, the newly revised version of the paper, $p_{rev}$, is re-evaluated using $\mathcal{R}^*$. This provides an updated feedback and overall score for the revised paper, $S_{rev}$.

4. **Select & Refine:** The overall score for the revised paper is logged, and the search process continues by selecting the current best scoring version of the paper which is fed back to the revision step again in attempt to further improve the paper.

Similarly, for the **Select & Refine** step, we utilize the MultiAIDE approach for implementation and name this search as MultiAIDE-Revision. In the **Revision** step, the model modifies the text of the paper, it is prompt specifically not to alter any of the experimental results and only perform changes to the presentation of the paper. Thus this process is able to start from an original paper and the discovered rubric to generating a revised version of the paper that maximizes the score from the scoring rubric.

## 4 EXPERIMENTS

In this section, we present the results for predicting a paper's future impact via citation counts and for revising a paper's content based on the LLM generated reviews. Additional results comparing the consistency of LLM to human reviewers are provided in Sec. I.

**Dataset.** Our study is grounded in a large-scale dataset comprising of publicly available papers and their corresponding peer reviews from four recent, top-tier machine learning conferences: the International Conference on Learning Representations (ICLR) in 2024 and 2025, and the Conference on Neural Information Processing Systems (NeurIPS) in 2023 and 2024. This collection provides a comprehensive and contemporary snapshot of the scientific review process in a fast-moving field, encompassing a wide range of research topics, paper qualities, and review styles. Tab. 2 outlines the different paper types in our dataset. The percentage of accepted pa-

per is higher than the actual conference acceptance rate due to NeurIPS not making all submissions public as only a subset of rejected papers are opted-in to be made public by their authors.

To establish a proxy measure for a paper's future 'impact', we gathered citation data from Semantic Scholar using their 'influential citation' count data for our analysis (Valenzuela et al., 2015). This metric is designed to identify citations that significantly engage with the cited work, filtering out passing mentions. Thus it can be used to evaluate whether a paper has general popularity or deeper scientific influence. Our dataset includes 26,707 papers in total, with an average citation count of 2.07 (see Fig. A1). We use a 80% / 10% / 10% split for the train, validation, and test set split.

Table 1: Paper types in our dataset.

| Category | # Paper | Percentage |
|----------|---------|------------|
| Oral | 427 | 1.6% |
| Spotlight | 1,451 | 5.4% |
| Poster | 11,344 | 42.4% |
| Reject | 8,916 | 33.4% |
| Withdrawn | 4,569 | 17.1% |
| Total | 26,707 | 100% |

### 4.1 IMPACT PREDICTION

The goal is to establish a metric which reliably reflects the impact of a paper being reviewed, so it can be used for improving the paper. This is achieved using the approach outlined in Sec. 3.1.

**Implementation Details and Baselines.** In Sec 3.1, we defined the set of hyperparameters for our primary method. In our experiments here, we set the initial branch factor and branch factor to three, the debug probability to 0.5, and the debug depth to maximum five. The *Human scores baseline* uses the original human reviewer scores as input features to the negative binomial regression model. The *Average citation baseline* predicts the average citation for all inputs. The *MLP on paper embedding* baseline uses the SPECTER embedding model (Cohan et al., 2020b) as an input feature and learns an MLP with a $\ell_2$ loss to predict the citation counts. SPECTER is trained in a citation-informed way and thus provides a reliable feature for predict the citation counts. The *Paper embedding + PCA* baseline uses the same SPECTER embedding model but first performs principal component analysis (PCA) on the embeddings and then uses the projected PCs as input features to the regression model. Another agentic baseline is *Prompt breeder* (Fernando et al., 2024) which evolves the rubric used for scoring papers as a prompt in a self-improving manner. We set the maximum number of iterations for iterative methods to 200 steps to allow a comprehensive exploration of the solution space. We use grid search to search for the best hyperparameters for the *MLP on paper embedding*, and we also searched for the best number of PCs to use in the *Paper embedding + PCA* baseline. For the negative binomial regression model, we performed a grid search for the best hyperparameters. APRES and *Prompt breeder* rely on LLMs, thus we compared four different frontier LLMs: o1, o3, Gemini 2.5 Flash, and Gemini 2.5 Pro. Dynamic thinking is turned on for Gemini models.

**Citation Number Prediction Results.** Results are presented in Fig. 3. Each experiment is repeated three times to generate the plotted confidence intervals. Our MultiAIDE-Rubric search approach demonstrates strong performance. Within just a few iterations, it discovers a rubric that yields a significantly lower MAE than all other methods, quickly converging to a stable, superior performance level of below 2.0 MAE for Gemini 2.5 Pro. This stands in contrast to the tested baselines. *Human scores baseline*, which uses the numerical scores from the original human reviewers, performs the worst, with an MAE near 5.0. This performance is very close to the *Average citation baseline* of MAE 5.3, indicating that the raw scores provided by humans are a relatively poor predictor of future citations. The two baselines using SPECTER embeddings (Cohan et al., 2020b) perform better, with the *MLP on Paper embedding* and the best *Paper embedding + PCA* model achieving MAEs of approximately 2.8 and 2.65, respectively. *Prompt breeder* (Fernando et al., 2024) shows initial promise but generally converges to a worse MAE than MultiAIDE-Rubric.

Comparing across base LLM models, the MultiAIDE-Rubric search consistently achieves the lowest MAE score while the OpenAI models o1 and o3 achieve lower MAE scores of 2.25 and 1.92, respectively. Gemini 2.5 Pro model is competitive with a MAE score of 1.96, Gemini 2.5 Flash shows a slightly higher MAE of 2.30. The trend of the search dynamics is also clear, the model first quickly converges to a low MAE score in the first 25 steps of search, and then slowly explores the solution space and roughly converges to the final solution after 100 steps. The key takeaway from this experiment is that the rubric discovered through our MultiAIDE-Rubric search are more predictive of future citations than features derived from the paper's content via embedding

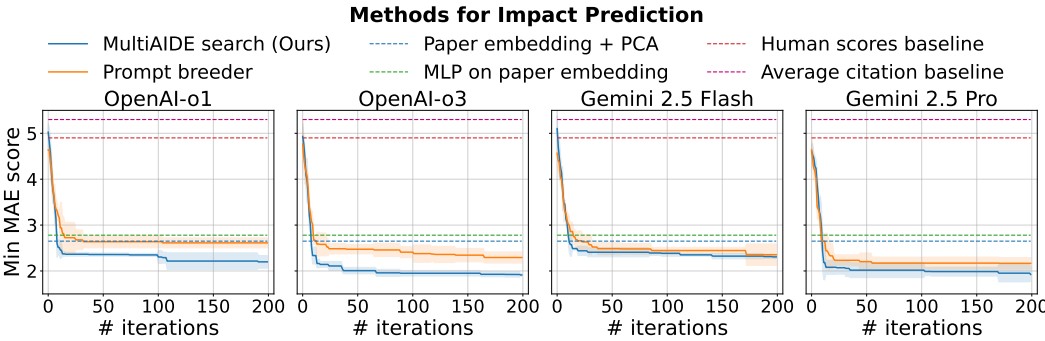

Figure 3: Performance of agentic search (MultiAIDE-Rubric) in predicting citation counts, measured in Mean Absolute Error (MAE). Our MultiAIDE-Rubric search approach converges to a lower MAE compared to several strong baselines, including a model using human reviewer scores (Human scores baseline), an MLP on SPECTER paper embeddings, a negative binomial model on the principal components of SPECTER embeddings (Paper embedding + PCA), and an alternative search method (Prompt breeder). The x-axis represents the number of iterations for search methods.

models or the scores assigned by human experts. This suggests an iterative search guided LLM can identify and quantify signals of future impact that are not captured by existing methods.

## 4.2 Paper Improvement

In this section, we evaluate the performance of our proposed paper revision approach from Sec. 3.2 that improves a paper's text with the aim of enhancing impact and human readability.

**Implementation Details and Baselines.** We adopted a diff-based editing method when editing the paper. Specifically, instead of prompting the 'Rewriter' to output the entire revised paper after reading the original one, we prompt it to specify edits as a series of search/replace blocks. In the search blocks, the model outputs the exact same content as in the part of the original paper it wants to edit, and in the replace block, the model outputs the content after its edit using the diff format from Aider. This diff-based editing method prevents cases where the 'Rewriter' tends to drastically shorten the original paper when trying to output the whole revised paper as each edit will only change a local part of the paper. Additionally, we can easily track where the 'Rewriter' wants to edit and prevent editing the experimental results directly. Specifically, we constraint the edits to be focused on the paper's text, excluding the tables where the experimental results are presented. We achieve this by encoding such restrictions in the prompt as well as preventing edits from applying to tables. For the search, we use the same hyperparameters as in Sec 4.1. We set the maximum number of iterations to 120 due to the total time cost for running the experiments. We compare to two baselines, *Simple Rubric* uses a simple rubric encoding clarity, correctness, contribution and impact, and presentation and style for guiding the paper revision process and *Embedding PCA* use the numerical feedback from the *Paper embedding + PCA* baseline for predicting impact.

**Evaluation Metrics.** The success of this paper improvement process is measured by two key metrics. The first metric is the *Improvement Score ($\Delta S$)*, which is the change between the final and original predicted impact scores, $\Delta S = S_{\text{rev}} - S_{\text{ori}}$. The impact scores are predicted by the discovered rubric in Sec. 3.1 as they correlate the most with the actual citation numbers. A positive $\Delta S$ provides quantitative evidence that the LLM-generated feedback was sufficiently constructive and actionable to guide the agent to produce a verifiably better paper. For the second metric, we conducted a user study with annotators who have PhDs in machine learning research and engineering who read and compared two versions of the same paper and provided a preference and justification.

As our paper revision approach can only improve the given paper by changing how the method is presented or how the contribution is described without any access to an environment that allows it to rerun experiments, it cannot change how the proposed method in the given paper performs. Therefore, for papers that were rejected because the proposed method does not perform well compared to other methods, the improvement from APRES can be small. While at the same time, papers that were rejected because the writing style prevented the reviewer from understanding and recognizing the impact of the work can potentially benefit the most from our approach. Thus in our evaluation, we not only consider the overall averaged improvement, but also the improvement on papers that are in different brackets: 'Clear Reject' papers with human reviewers scores far below the averaged

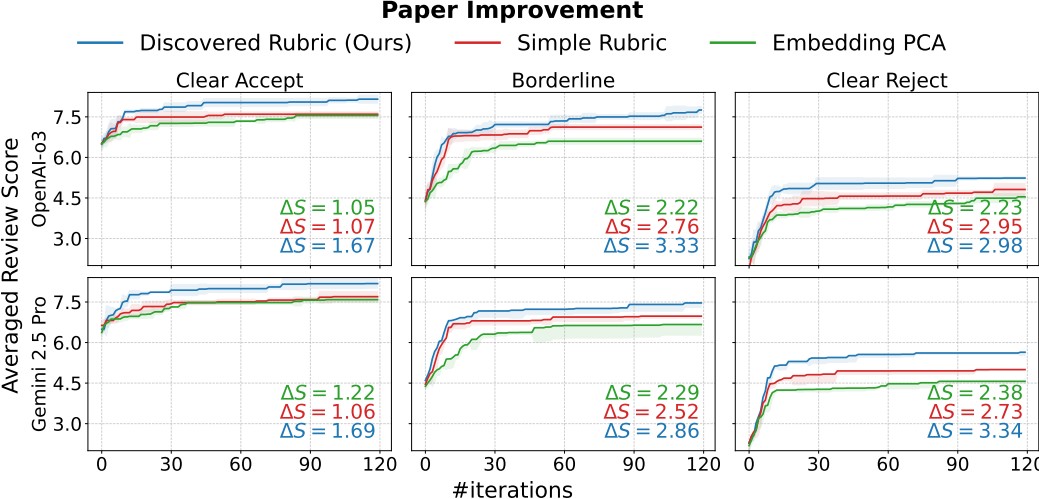

Figure 4: Improvement in predicted paper quality from our iterative revision process using `OpenAI-o3` and `Gemini 2.5 Pro` LLMs. Our agentic 'Rewriter' improves the scores for both borderline and clear reject papers more. The higher final scores obtained for borderline papers suggests our rubric discovery approach is more effective at fixing presentational flaws as opposed to fundamental scientific ones. The y-axis is averaging of the scores from all rubric items using the same discovered rubrics in stage 1. $\Delta S$ stands for the change between the final and original predicted impact scores.

scores of all papers, 'Borderline' papers that are close to the borderline, and 'Clear Accept' papers that received the highest scores from human reviewers.

**LLM Rubric Evaluation Results.** To evaluate the utility of LLM feedback, we applied our agent-based revision approach to the test set. The results of this iterative process are shown in Fig. 4. The primary finding is that the 'Rewriter' successfully improves the predicted impact score for all categories of papers, as evidenced by the increasing scores for all configurations. This demonstrates that LLM-generated feedback is indeed actionable and can help to make concrete improvements.

Furthermore, the results reveals two crucial insights. First, using the 'Discovered Rubric' as the objective function for revision leads to significantly higher final scores than using the *Embedding PCA* baseline. This reinforces the insight from our previous experiment that the discovered rubric is potentially a more effective measure of paper quality. Second, we observe a distinct difference in the improvement trajectory between borderline and rejected papers. While both groups benefit from revision, the 'borderline' papers start from a higher baseline and achieve a much higher final score with $\Delta S = 3.33$ for `o3` model and $\Delta S = 2.86$ for `Gemini 2.5 Pro`, pushing them firmly into what might be considered acceptance territory. In contrast, the 'Clear Reject' papers, while improved, plateau at a substantially lower quality ceiling. The $\Delta S$ is 2.98 and 3.34 for `o3` and `Gemini 2.5 Pro` respectively, similar to the improvement on the 'borderline' papers. This suggests that our automated revision process is most effective at correcting flaws related to presentation, clarity, or argumentation—the kinds of issues that might land an otherwise solid paper in the borderline category. In contrast, it is less effective at fixing deep fundamental problems with a paper's core method, contribution, or results, which are more likely to be the cause of an initial rejection.

Across the `o3` and `Gemini 2.5 Pro` models, the $\Delta S$ metric on three categories of papers remains similar. Notably, for 'Clear Accept' papers, the $\Delta S$ metric is much smaller than the $\Delta S$ for the other two categories, suggesting that paper quality saturates. The trends for the paper quality improvements are also similar across models. These improvements are notable, as we are improving top-tier conference papers, most of them already have a high text quality prior to submission.

Table 2: Percentage of human participant preferences for the revised papers.

| Category | # Papers | Percentage |
|---|---|---|
| 3/3 participants preferred | 174 | 47.8% |
| 2/3 participants preferred | 113 | 31.0% |
| 1/3 participants preferred | 65 | 17.9% |
| 0/3 participants preferred | 12 | 3.3% |
| Total | 364 | 100% |

**Human Evaluation of Revised Papers.** We further evaluated the agentic revised papers by having human participants provide pairwise preference evaluation for them (see Tab. 2). The human

participants were researchers and engineers with rich experience in machine learning, and all have relevant PhD degrees or in the process of obtaining one. In each evaluation round, an annotator was presented with a pair of papers (the original paper and the version after AI revision) and were asked to determine which one they found to be of higher quality. The pairwise evaluation was done in a blind fashion, where the participants did not know which paper of the pair was from the AI revision. In total, we obtained ratings for 364 pairs of papers, where each pair was evaluated by three different participants. Out of the 364 pairs of papers, 287 paper are preferred by the majority of the human participants. A binomial test rejects the null hypothesis of equal preference with a p-value of less than $10^{-22}$. The corresponding 95% confidence interval for the probability of the revised paper being preferred lies between 70.1% and 79.0%, indicating strong performance. Fig. 5 shows common reasons of the participants in justifying the preference towards the revised papers.

**Influence of Discovered Rubric and Search.** APRES relies on two components, the discovered rubric from Sec. 3.1 and the iterative search. Here we ablate the influence of the two components on the performance. Removing the discovered rubric means that the agent will not be prompted with the rubric but with just a goal of improving the potential impact of the paper. In Tab. 3 we observe that removing either the discovered rubric $\mathcal{R}^*$ or the MultiAIDE-Revision search process greatly hurts the performance.

Table 3: Ablation of APRES components.

| Paper Type | 'Accept' | 'Borderline' | 'Reject' |
|---|---|---|---|
| APRES w/o $\mathcal{R}^*$ | 1.02 | 1.24 | 1.21 |
| APRES w/o MultiAIDE-Revision | 1.13 | 1.46 | 1.34 |
| APRES | 1.67 | 3.33 | 2.98 |

## 5 DISCUSSION

**From Mimicking Humans to Predicting Impact.** Comparing the limited success of LLMs in predicting impact when using human-like review guidelines with the success of APRES suggests the primary value of LLMs may not be in mimicking human reviewers, but in discovering novel, more effective evaluation criteria. In this work, we assume that citation numbers are a reasonable proxy for the impact of a paper. However, this assumption is subject to many biases, such as the number of researchers working on a given topic, the timing of a paper's release, and the promotion of a work on social media. Moreover, as with any quantifiable metric, when a measure becomes a target it ceases to be a good measure (Goodhart's Law) (Strathern, 1997), it is likely to be gamed and may fail to reflect the true impact of a paper as both human authors and AI systems optimize for it. Quantitative measures will likely need to be complemented with adaptive approaches or alternative ways of capturing scientific impact.

**The Path to More Helpful Reviews.** Our paper improvement experiments suggest that while LLM feedback can lead to improvements, the focus on surface-level aspects like writing style indicates a need for developing methods to elicit deeper, more constructive technical feedback. Analysis by Goldberg et al. (2024) shows that iterative deep constructive technical feedback also leads to fewer authors submitting their work and results in higher frustration from authors. Therefore, LLM-based systems for paper revision should consider more human-in-the-loop interactions.

**Reviews and Verifiable Reproducibility.** While the preceding discussion has focused on the inherently unverifiable aspects of human peer review, where subjectivity and inconsistency can take place, there exists a parallel, verifiable domain that is increasingly gaining prominence (Black et al., 2025; Xiang et al., 2025). This domain centers on a paper's adherence to objective criteria such as the authors' checklist, the availability and clarity of codebases or pseudocode, and the inclusion of sufficient methodological details to enable independent reproduction of results. The development of robust mechanisms for automatic checking of a paper's reproducibility

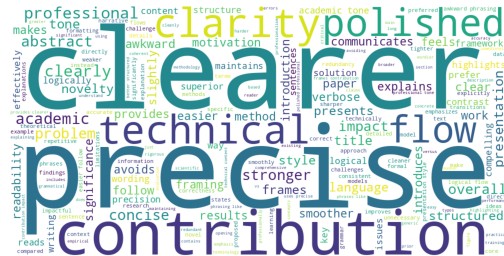

Figure 5: The word cloud generated from participants' reasons for preferring a revised paper.

represents a complementary effort to improving the quality and consistency of reviews, having the potential to significantly enhance the integrity and reliability of the scientific publication process.

**Application to Other Domains.** Our experiments are conducted on AI conferences (i.e., ICLR and NeurIPS) primarily due to data availability. However, the core methodology we propose is domain-

agnostic. For example, it could be applied to a Biology corpus with citation data, and would simply discover a different relevant rubric.

**Limitations.** Our analysis is based exclusively on the textual content of the papers. APRES does not process figures, which are often critical for conveying key results and understanding a paper's contribution. Additionally, while our paper revision agent was explicitly prompted to only improve the presentation without altering the scientific content, we cannot perfectly guarantee this constraint was met. The LLM could have inadvertently introduced subtle inaccuracies. For instance, by altering a technical description of the methodology in a way that no longer reflects the original experiment. Future work should incorporate multi-modal analysis and develop more robust verification methods to ensure the scientific integrity of automated revisions. Finally, adversarial attacks on LLMs have also been shown to influence the judgment of LLM reviewers (Collu et al., 2025). While we do not explore the influence of such attacks, future automatic approaches for paper reviewing and improvement should take this into account.

## 6 CONCLUSION

We introduced APRES, a new approach for automatically improving the clarity and readability of a scientific paper by leveraging a rubric discovered by maximizing a proxy for scientific impact. We showed that our iterative approach can successfully revise existing papers to improve their predicted citation count, which we verify with both automated and human evaluation. Our findings provide strong evidence for using such methods as tools to help authors "stress-test" their manuscripts and improve the clarity of their text. We do not advocate for replacing human experts, but rather for augmenting the peer review ecosystem with sophisticated AI agents that reflect human values. In this way, AI can serve as a catalyst to help humans communicate scientific discoveries more clearly, thereby unlocking faster, safer, and more impactful scientific progress.

ETHICS STATEMENT

This work introduces APRES, a method that (i) searches for a review rubric predictive of future impact and (ii) uses that rubric to suggest text edits that improve clarity without altering scientific claims. The goal is to support authors and augment, not replace, expert peer review. We align our practices with the ICLR Code of Ethics and related community norms on research integrity, fairness, transparency, and the avoidance of harm.

**Data sources and licensing.** We use publicly available manuscripts and reviews from recent ICLR and NeurIPS editions hosted on OpenReview, and citation counts from Semantic Scholar. We accessed only content made public by the venues/authors under their terms of use. We did not scrape private materials, attempt any deanonymization, or circumvent access controls. When releasing artifacts, we will share code, prompts, and derived, non-identifying labels/metrics. We will not redistribute third-party PDFs or reviews beyond what the original platforms already make public, and we will respect original licenses.

**Human subjects.** Our human evaluation involved adult annotators with AI experience who provided pairwise preferences between an original paper and its AI-revised counterpart in a blind setup. Our human evaluators were recruited via specialized, compliant public online platforms (e.g., Outlier, Turing). These platforms ensured that all annotators met the necessary professional expertise, specifically requiring them to be PhD-level researchers (or actively pursuing one) in machine learning. Utilizing these platforms confirms that all annotators met the stated conditions and that the process conforms to relevant regulations for human annotation studies, ensuring the integrity and high standards of our pairwise preference evaluation. No personally identifiable information was collected. Annotators gave informed consent and were compensated at or above local fair hourly rates. The study involved minimal risk and no intervention.

**Risk of harm and dual use.** A system that predicts impact or revises text could be misused to: 'game' perceived impact (e.g., optimizing for citation-like signals), unduly influence reviewers, homogenize scientific writing or disadvantage out-of-distribution research communities. We take the following precautions: (i) *Constraint on edits:* the rewriter is restricted to presentation (organization, clarity, and exposition) and is explicitly instructed not to change technical claims, reported results, or tables. (ii) *Intended use and disclosure:* we position APRES as a pre-submission authoring assistant and recommend disclosure when AI assistance materially impacts a manuscript. (iii) *Evaluation framing:* the discovered rubric is a surrogate objective. We caution against treating it as a definitive measure of quality. Human experts remain the ultimate arbiters of scientific merit. (iv) *Misuse monitoring:* we do not provide mechanisms to target specific venues, reviewers, or to manipulate peer-review platforms, and we will monitor and address reported misuse.

**Fairness, bias, and generality.** Using citations as a proxy can encode field-, venue-, and trend-specific biases (e.g., larger subfields, English-language advantage, topical popularity). There is a possibility that our rubric search could inherit these biases. We partially mitigate this by: (i) reporting performance across paper strata (e.g., accepted/borderline/rejected), (ii) analyzing failure cases and distributional shifts, (iii) encouraging community replication on alternative objectives (e.g., expert quality ratings, reproducibility checklists). We do not filter by author identity or demographics and we do not infer such attributes.

**Privacy and confidentiality.** We process only public artifacts. No private submission materials or confidential reviewer identities are used. For human studies, we store only anonymized preferences and free-text rationales. No raw platform access tokens or personal data are shared.

**Research integrity and transparency.** We avoid fabrication or alteration of scientific content. All model prompts, hyperparameters, selection criteria, and ablations are documented to enable reproducibility.

REPRODUCIBILITY STATEMENT

We have made extensive efforts to ensure reproducibility. All implementation details of APRES, including model prompts, search hyperparameters, and evaluation pipelines, are documented in the main paper and appendix. The datasets used are drawn from publicly available sources (OpenReview submissions and Semantic Scholar citation counts), and our preprocessing steps are described in the appendix. To further facilitate replication, we will release a repository containing the full codebase, instructions, derived data splits, and evaluation scripts upon publication.

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

# Appendix

## A    LLM Use

Although the method proposed in this work is an approach for revising research papers with LLMs, we did not run the proposed method or any other LLM system for writing assistance on this paper.

## B    Comparison to Related Work and Dataset Information

In Tab. A1 we compare our APRES approach to the existing methods in the literature. In Fig. A1 we display a histogram of the citation counts of the papers used in our dataset. See Sec. 4 for more information about our dataset.

In the experiments in the main paper and Fig. A3 and A2, we classified papers into three categories, 'Clear Accept', 'Borderline', and 'Clear Reject'. For ICLR, the borderline review score is 5, and for NeurIPS the borderline review score is 4. Therefore, we classify papers with average review scores from human reviewer higher than 6 for ICLR and 5 for NeurIPS as 'Clear Accept', and scores lower than 4 for ICLR and 3 for NeurIPS as 'Clear Reject', and the rest as 'Borderline'.

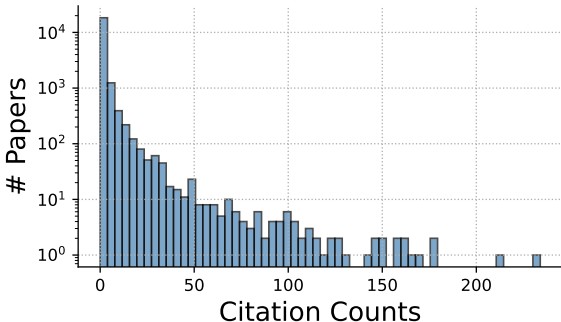

Figure A1: Distribution of citations in our dataset.

Table A1: Comparison of our work to related approaches. APRES is the first method to integrate the discovery of predictive evaluation criteria with a closed-loop system for automated paper revision.

| Approach | LLM Driven | Predicts Future Impact | Discovers Predictive Criteria | Guides Automated Revision |
|---|:---:|:---:|:---:|:---:|
| ReviewRobot (Wang et al., 2020) | ✗ | ✗ | ✗ | ✗ |
| Reviewer 2 (Gao et al., 2024) | ✓ | ✗ | ✗ | ✗ |
| MARG (D'Arcy et al., 2024) | ✓ | ✗ | ✗ | ✗ |
| ReviewRL (Zeng et al., 2025) | ✓ | ✗ | ✗ | ✗ |
| DGNI (Geng et al., 2022) | ✗ | ✓ | ✗ | ✗ |
| TNCSI$_{SP}$ (Zhao et al., 2025b) | ✓ | ✓ | ✗ | ✗ |
| HLM-Cite (Hao et al., 2024) | ✓ | ✓ | ✗ | ✗ |
| PEGASUS (Zhang et al., 2019) | ✗ | ✗ | ✗ | ✓ |
| SWIF$^2$T (Chamoun et al., 2024) | ✓ | ✗ | ✗ | ✓ |
| R3 (Du et al., 2022) | ✓ | ✗ | ✗ | ✓ |
| **APRES** (Ours) | ✓ | ✓ | ✓ | ✓ |

## C    PDF Parsing and Content Preservation

We downloaded all paper pdfs from OpenReview[1], and then processed all pdfs with the SciPDF parser[2] to obtain the text for all papers. Then the papers were processed like text files for APRES. The SciPDF parser is able to parse the main texts and the figures and tables to different sections, therefore we can programmatically control which part the APRES edits are applied to. This serves

---

[1] https://openreview.net/
[2] https://github.com/titipata/scipdf_parser

as a deterministic code-based filter to prevent edits to tables and figures. As SciPDF is directly using the PDF metadata for parsing, the process of labeling contents as figures or tables is accurate. After the parsing of SciPDF, we lock all text contents classified as tables, and as a result they cannot be updated during the revision process of APRES. Additionally, we did not include the parsed author information into the text to prevent information leakage.

To test that this process for content preservation, we run this process on 500 papers and then manually checked all proposed changes made by APRES. In all of the cases examined, none of the scientific results were changed by APRES. This demonstrates that this guardrail is able to block any proposed changes to table contents.

## D    PAPER IMPROVEMENT USING OTHER LLMS

In Fig. A2 and A3, we present the results for paper improvement using `o1` and `Gemini 2.5 Flash`. The general trend of the revision improvement is consistent to what we showed in the main paper (see Fig. 4).

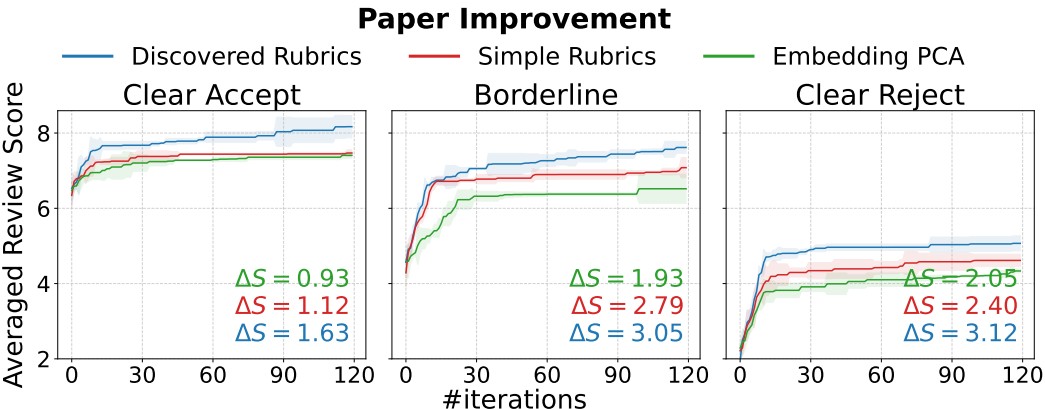

Figure A2: Improvement in predicted paper quality during the iterative revision process using the `o1` model. The agentic rewriter improves the scores for both 'borderline' (solid lines) and 'rejected' (dashed lines) papers. The higher final scores achieved for borderline papers suggests our approach is more effective at fixing presentational flaws than fundamental scientific ones.

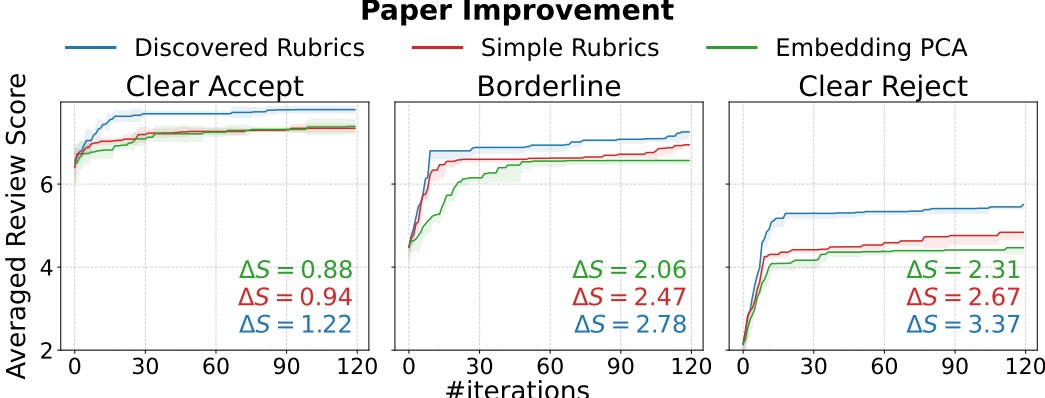

Figure A3: Improvement in predicted paper quality during the iterative revision process using the `Gemini 2.5 Flash` model. The agentic rewriter improves the scores for both 'borderline' (solid lines) and 'rejected' (dashed lines) papers. The higher final scores achieved for borderline papers suggests our approach is more effective at fixing presentational flaws as opposed to fundamental scientific ones.

# E  DISCOVERED RUBRIC

Here we present the discovered paper reviewing rubric obtained from APRES with `OpenAI-o1` as the base LLM. The rubric includes over 60 items, and covers aspects like "Problem Formulation & Significance", "Literature Review & Context", "Methodology & Technical Rigor", "Results & Analysis", "Discussion & Conclusion", "Originality & Novelty", "Writing & Presentation Quality", and "Future Impact & Influence". Note that all texts in the box below is discovered by APRES, including the comments.

---

**Discovered Rubric**

```
EVALUATION_RUBRIC = {
    # I. Problem Formulation & Significance
    "problem_clarity": "Is the problem statement unambiguous, well-
        defined, and easy to understand?",
    "field_relevance": "Does the problem address a significant and
        recognized challenge or gap
within its primary field?",
    "timeliness": "Is the problem currently relevant and of high
        interest to the research
community?",
    "problem_scope": "Is the scope appropriately scoped-not too
        broad to be intractable, nor
too narrow to be trivial?",
    "motivation": "Is there a compelling justification for why this
        problem needs to be solved?",
    "hypothesis_clarity": "Are the central research questions or
        hypotheses explicitly stated
and well-formed?",
    "impact_potential_of_problem": "Does solving this problem have
        the potential for
transformative impact on theory or practice?",

    # II. Literature Review & Context
    "lit_review_comprehensiveness": "Does the literature review
        cover the most relevant and
seminal prior work?",
    "lit_review_accuracy": "Is prior work represented accurately and
         fairly, without straw-manning?",
    "gap_identification": "Does the review clearly identify a
        specific, demonstrable gap that
the current work aims to fill?",
    "critical_analysis_of_lit": "Does the paper critically analyze
        previous work rather than
just summarizing it?",
    "recency_of_references": "Are the references up-to-date,
        reflecting the current state of the art?",
    "positioning_vs_alternatives": "Is the work effectively
        differentiated from and compared to
the closest existing approaches?",

    # III. Methodology & Technical Rigor
    "method_appropriateness": "Is the chosen methodology suitable
        for addressing the research
questions?",
    "method_description_detail": "Is the methodology described with
        sufficient detail to allow
for replication?",
    "justification_of_choices": "Are the choices of methods,
        parameters, and configurations
well-justified?",
```

---

```
"technical_correctness": "Is the application of the methodology
    free from technical,
mathematical, or logical errors?",
"assumptions_clarity": "Are all key assumptions explicitly
    stated and their validity
discussed?",
"data_quality": "Is the data (or corpus) used of high quality
    and appropriate for the
problem?",
"baselines_and_comparisons": "Are the chosen baselines or
    control groups strong,
relevant, and fairly compared?",
"ethical_considerations": "Are potential ethical issues related
    to the data
or methodology addressed?",
"method_limitations": "Are the inherent limitations of the
    chosen methodology acknowledged?",

# IV. Results & Analysis
"clarity_of_results_presentation": "Are the results presented
    clearly through
well-designed tables, figures, and text?",
"support_for_claims": "Do the presented results directly and
    convincingly support
the paper's central claims?",
"depth_of_analysis": "Does the analysis go beyond superficial
    observations to provide
deeper insights?",
"statistical_validity": "Is the statistical analysis (if any)
    appropriate, correctly
executed, and properly interpreted?",
"robustness_and_sensitivity": "Have the results been tested for
    robustness
(e.g., via sensitivity analysis, different datasets)?",
"reporting_of_negative_results": "Are negative or unexpected
    results reported
and analyzed honestly?",
"interpretation_soundness": "Is the interpretation of the
    results logical, cautious,
and free from overstatement?",
"result_significance": "Are the results not only statistically
    significant but also
practically or theoretically meaningful?",

# V. Discussion & Conclusion
"synthesis_of_findings": "Does the discussion effectively
    synthesize the results
and connect them back to the initial research questions?",
"contribution_articulation": "Is the paper's core contribution
    stated clearly
and concisely in the conclusion?",
"implications_discussion": "Are the broader theoretical and/or
    practical
implications of the findings discussed thoughtfully?",
"generalizability_discussion": "Is there a reasoned discussion
    on the generality
and boundaries of the findings?",
"study_limitations_discussion": "Does the paper offer a candid
    discussion of
its overall limitations (beyond just methodology)?",
"future_work_suggestions": "Are the suggestions for future work
    specific,
insightful, and non-obvious?",
```

```
# VI. Originality & Novelty
"conceptual_novelty": "Does the paper introduce a new concept,
    framework, or perspective?",
"methodological_novelty": "Does it propose a new algorithm,
    technique, or procedure?",
"empirical_novelty": "Does it present novel empirical findings
    or a valuable new dataset?",
"synthetical_novelty": "Does it connect previously disconnected
    ideas in a new and insightful way?",
"paradigm_shift_potential": "Does the work have the potential to
     challenge
established assumptions or shift the direction of the field?",
"advance_over_sota": "Is the contribution a significant advance
    over the state-of-the-art,
or merely an incremental improvement?",

# VII. Writing & Presentation Quality
"title_and_abstract_quality": "Does the title effectively
    capture the core topic?
Does the abstract provide an accurate, comprehensive summary?",
"logical_structure_and_flow": "Is the paper well-organized with
    a logical and intuitive flow?",
"clarity_of_language": "Is the writing clear, precise, and
    unambiguous?",
"conciseness": "Is the paper written concisely, without
    unnecessary jargon or verbosity?",
"grammar_and_style": "Is the manuscript free of grammatical,
    spelling, and stylistic errors?",
"figure_and_table_quality": "Are figures and tables high-quality
    , well-labeled,
and easy to interpret?",

# VIII. Future Impact & Influence
"aha_factor": "Does the paper provide a fundamental insight that
     changes
how one thinks about a problem?",
"foundation_for_future_work": "Is the work likely to be
    frequently cited and built
upon by others?",
"interdisciplinary_appeal": "Does the work have the potential to
     influence or be
adopted by other fields?",
"scalability": "Is the proposed idea or method scalable to more
    complex, real-world scenarios?",
"simplicity_and_elegance": "Is the core idea surprisingly simple
    , elegant, and powerful?",
"resource_provision": "Does the paper provide valuable community
     resources
(e.g., open-source code, datasets, benchmarks)?",
"educational_value": "Could this paper serve as an excellent
    pedagogical tool for
teaching a key concept?",
"debate_provocation": "Is the paper likely to spark important
    new discussions or
productive controversies?",
"practical_applicability": "Does the work have clear potential
    for real-world application
or commercialization?",
"societal_impact": "Are the potential positive or negative
    societal consequences of
the work significant?",
```

```
    "memorable_takeaway": "Is there a clear, impactful, and
        memorable 'take-home message'?",
    "efficiency_improvement": "Does the contribution offer a
        substantial improvement in efficiency,
    cost, or performance over existing solutions?",
    "opens_new_directions": "Does the work open up entirely new
        avenues of research?"
}

SCORING_GUIDELINES = {
    # I. Problem Formulation & Significance
    "problem_clarity": {
        0: "Problem statement is ambiguous, confusing, or completely
            unintelligible.",
        5: "Problem is generally understandable but lacks precision
            or contains some ambiguity.",
        10: "Problem statement is exceptionally clear, precise, and
            unambiguous."
    },
    "field_relevance": {
        0: "Addresses a trivial, irrelevant, or non-existent problem
            for the field.",
        5: "Addresses a recognized problem, but its importance or
            relevance is minor.",
        10: "Addresses a critical, high-priority, and widely
            recognized challenge in the field."
    },
    "timeliness": {
        0: "The problem is outdated, obsolete, or has been superseded
             by other work.",
        5: "The problem is relevant but is not a current area of high
             interest or activity.",
        10: "The problem is highly timely and addresses a pressing,
            current topic of interest."
    },
    "problem_scope": {
        0: "The scope is intractably broad or so narrow that it
            becomes trivial.",
        5: "The scope is reasonable but could be better defined,
            leading to some unfocused aspects.",
        10: "The scope is perfectly defined to allow for a deep,
            meaningful,
        and self-contained contribution."
    },
    "motivation": {
        0: "The paper provides no compelling justification for why
            the work is needed.",
        5: "The motivation is present but is not strongly argued or
            is based on weak assumptions.",
        10: "The paper provides a powerful, convincing, and well-
            supported motivation for the work."
    },
    "hypothesis_clarity": {
        0: "No clear research questions or hypotheses are stated or
            can be inferred.",
        5: "Research questions or hypotheses are implied or stated
            vaguely.",
        10: "The paper explicitly states clear, specific, and
            testable research questions or hypotheses."
    },
    "impact_potential_of_problem": {
        0: "Solving this problem would have no meaningful impact on
            the field.",
```

```
        5: "Solving this problem would represent a minor, incremental
            improvement.",
        10: "Solving this problem would have a transformative or
            groundbreaking impact on the field."
    },

    # II. Literature Review & Context
    "lit_review_comprehensiveness": {
        0: "Completely misses key literature and seminal works in the
            area.",
        5: "Covers most of the relevant work but has some noticeable
            omissions.",
        10: "Provides a comprehensive, thorough, and complete survey
            of all relevant literature."
    },
    "lit_review_accuracy": {
        0: "Grossly misrepresents or misunderstands prior work.",
        5: "Generally represents prior work accurately but with some
            minor misinterpretations.",
        10: "Represents all prior work with exceptional nuance,
            accuracy, and fairness."
    },
    "gap_identification": {
        0: "Fails to identify any gap in the existing literature.",
        5: "Identifies a general area for improvement but does not
            specify a clear gap.",
        10: "Clearly and precisely articulates a specific,
            demonstrable gap the work addresses."
    },
    "critical_analysis_of_lit": {
        0: "The literature review is a simple list or summary with no
            analysis.",
        5: "Includes some light analysis but is still mostly
            descriptive.",
        10: "Provides a deep, critical analysis of the literature,
            synthesizing it effectively."
    },
    "recency_of_references": {
        0: "Relies entirely on outdated sources, missing recent and
            key advances.",
        5: "Includes some recent work but misses the absolute state-
            of-the-art.",
        10: "The references are fully up-to-date with the very latest
            , cutting-edge research."
    },
    "positioning_vs_alternatives": {
        0: "Fails to position the work or compare it to any
            alternatives.",
        5: "Provides a superficial comparison to some obvious
            alternatives.",
        10: "Provides a thorough and insightful comparison to the
            most relevant alternatives."
    },

    # III. Methodology & Technical Rigor
    "method_appropriateness": {
        0: "The methodology is entirely inappropriate for the
            research question.",
        5: "The methodology is acceptable but suboptimal; other
            methods would be more suitable.",
        10: "The methodology is perfectly suited for the research
            question and rigorously applied."
    },
```

```
"method_description_detail": {
    0: "Description is completely lacking in detail, making
        replication impossible.",
    5: "Key details are missing or ambiguous, making replication
        difficult.",
    10: "Method is described with sufficient detail for an expert
         to replicate the work."
},
"justification_of_choices": {
    0: "No justification is provided for methodological choices,
        parameters, or configurations.",
    5: "Some choices are justified, but many are left unexplained
        or poorly rationalized.",
    10: "All methodological choices are clearly explained and
         rigorously justified."
},
"technical_correctness": {
    0: "Contains severe technical, mathematical, or logical
        errors that invalidate the results.",
    5: "Contains minor technical errors or inaccuracies that do
        not invalidate the core claims.",
    10: "The methodology is technically flawless and correctly
         implemented."
},
"assumptions_clarity": {
    0: "Fails to state the key assumptions underlying the
        methodology.",
    5: "Some assumptions are stated, but others are implicit or
        their impact is not discussed.",
    10: "All key assumptions are explicitly stated and their
         potential impact is discussed."
},
"data_quality": {
    0: "Data is flawed, inappropriate, or of very poor quality.",
    5: "Data is adequate but has some limitations that are not
        fully addressed.",
    10: "The paper uses high-quality, appropriate, and well-
         vetted data for the problem."
},
"baselines_and_comparisons": {
    0: "No baselines are used, or the chosen baselines are
        trivial 'straw-man' comparisons.",
    5: "Baselines are relevant but are not state-of-the-art or
        are weakly implemented.",
    10: "The paper compares against strong, relevant, and state-
         of-the-art baselines."
},
"ethical_considerations": {
    0: "Ignores obvious and serious ethical issues.",
    5: "Briefly mentions ethical issues but with no substantive
        discussion.",
    10: "Provides a thoughtful and thorough discussion of all
         relevant ethical considerations."
},
"method_limitations": {
    0: "Fails to acknowledge any limitations of the chosen
        methodology.",
    5: "Acknowledges obvious limitations but does not discuss
        them in depth.",
    10: "Provides a candid and insightful discussion of the
         methodology's limitations."
},
```

```
        # IV. Results & Analysis
        "clarity_of_results_presentation": {
            0: "Results are presented in a confusing, misleading, or
                unintelligible manner.",
            5: "Results are understandable but poorly organized; figures/
                tables are unclear.",
            10: "Results are presented with exceptional clarity in well-
                designed figures, tables, and text."
        },
        "support_for_claims": {
            0: "The results do not support the paper's claims or directly
                contradict them.",
            5: "The results provide weak or indirect support for the
                paper's main claims.",
            10: "The results provide strong, direct, and convincing
                evidence for all claims."
        },
        "depth_of_analysis": {
            0: "Analysis is completely superficial, merely stating the
                results without interpretation.",
            5: "Analysis provides some basic interpretation but misses
                deeper insights.",
            10: "The analysis is deep, insightful, and uncovers non-
                obvious patterns or implications."
        },
        "statistical_validity": {
            0: "The statistical analysis is fundamentally flawed,
                inappropriate, or misinterpreted.",
            5: "The statistical analysis is mostly correct but has minor
                errors or questionable choices.",
            10: "The statistical analysis is rigorous, appropriate, and
                correctly interpreted."
        },
        "robustness_and_sensitivity": {
            0: "The robustness of the results is not tested or considered
                at all.",
            5: "A limited or superficial robustness check is performed.",
            10: "The paper includes a thorough analysis of the robustness
                and sensitivity of its results."
        },
        "reporting_of_negative_results": {
            0: "Likely negative or null results appear to be omitted or
                hidden.",
            5: "Negative results are mentioned briefly but not analyzed
                .",
            10: "Negative and unexpected results are reported
                transparently and analyzed for insights."
        },
        "interpretation_soundness": {
            0: "Interpretation is illogical, overstated, or completely
                disconnected from the results.",
            5: "Interpretation is plausible but slightly overstates the
                findings or ignores
            alternative explanations.",
            10: "Interpretation is cautious, well-reasoned, and fully
                supported by the results."
        },
        "result_significance": {
            0: "The results are trivial and have no practical or
                theoretical significance.",
            5: "The results are statistically significant but their
                practical/theoretical
            importance is minor.",
```

```
        10: "The results are highly significant, representing a major
             practical or theoretical advance."
    },

    # V. Discussion & Conclusion
    "synthesis_of_findings": {
        0: "The discussion fails to connect the results back to the
            research questions.",
        5: "The discussion recaps the results but provides little
            synthesis or integration.",
        10: "The discussion masterfully synthesizes the findings and
            relates them to the core questions."
    },
    "contribution_articulation": {
        0: "The main contribution is not stated or is completely
            unclear.",
        5: "The contribution is stated, but imprecisely or its
            importance is not clear.",
        10: "The paper's core contribution is articulated with
            exceptional clarity and precision."
    },
    "implications_discussion": {
        0: "No implications of the findings are discussed.",
        5: "Obvious implications are mentioned, but the discussion is
             superficial.",
        10: "Provides a thoughtful and insightful discussion of
            broader implications."
    },
    "generalizability_discussion": {
        0: "Makes sweeping claims about generalizability without
            evidence or discussion.",
        5: "Briefly discusses generalizability but without much depth
             or nuance.",
        10: "Provides a careful and well-reasoned discussion of the
            findings'
        generalizability and boundaries."
    },
    "study_limitations_discussion": {
        0: "Fails to acknowledge any of the study's overall
            limitations.",
        5: "Acknowledges only superficial limitations or dismisses
            them too quickly.",
        10: "Provides a candid, insightful, and self-critical
            discussion of the study's limitations."
    },
    "future_work_suggestions": {
        0: "Suggestions for future work are absent or completely
            generic (e.g., 'more work is needed').",
        5: "Suggests obvious next steps without providing much
            insight.",
        10: "Proposes specific, insightful, and non-obvious
            directions for future research."
    },

    # VI. Originality & Novelty
    "conceptual_novelty": {
        0: "The paper contains no new concepts, frameworks, or ideas
            .",
        5: "Offers a minor variation or combination of existing
            concepts.",
        10: "Introduces a genuinely new and powerful concept,
            framework, or perspective."
    },
```

```
      "methodological_novelty": {
          0: "Applies a standard, well-known methodology without
             modification.",
          5: "Proposes a minor or incremental modification to an
             existing technique.",
          10: "Develops a fundamentally new and impactful algorithm,
              technique, or procedure."
      },
      "empirical_novelty": {
          0: "Presents no new empirical findings or re-uses an old
             dataset without new insight.",
          5: "Presents new findings that are of limited scope or
             interest.",
          10: "Presents significant new empirical findings or a highly
              valuable new dataset."
      },
      "synthetical_novelty": {
          0: "The work is a simple application of existing ideas with
             no synthesis.",
          5: "Connects existing ideas in a straightforward or
             previously suggested way.",
          10: "Provides a novel and powerful synthesis of previously
              disconnected ideas."
      },
      "paradigm_shift_potential": {
          0: "The work firmly operates within and reinforces existing
             paradigms.",
          5: "The work gently pushes the boundaries of existing
             assumptions.",
          10: "The work fundamentally challenges established
              assumptions and
          has paradigm-shifting potential."
      },
      "advance_over_sota": {
          0: "The work does not improve upon or is worse than the state
             -of-the-art.",
          5: "Offers a minor, incremental improvement over the state-of
             -the-art.",
          10: "Represents a substantial, significant leap forward over
              the state-of-the-art."
      },

      # VII. Writing & Presentation Quality
      "title_and_abstract_quality": {
          0: "Title is uninformative or misleading; abstract is
             inaccurate or incomplete.",
          5: "Title and abstract are adequate but could be more
             informative or concise.",
          10: "Title is compelling and informative; abstract is an
              excellent, accurate summary."
      },
      "logical_structure_and_flow": {
          0: "The paper is disorganized and extremely difficult to
             follow.",
          5: "The structure is logical but transitions are sometimes
             abrupt or unclear.",
          10: "The paper is exceptionally well-organized with a clear,
              logical, and smooth flow."
      },
      "clarity_of_language": {
          0: "The writing is unclear, convoluted, and filled with
             ambiguity.",
```

```
        5: "The writing is generally clear but has some awkward
            phrasing or unclear sentences.",
        10: "The writing is exceptionally clear, precise, and a
            pleasure to read."
    },
    "conciseness": {
        0: "The paper is extremely verbose, repetitive, and filled
            with unnecessary jargon.",
        5: "The paper is somewhat wordy or could be more direct in
            its language.",
        10: "The paper is concise and communicates its points with
            admirable brevity."
    },
    "grammar_and_style": {
        0: "The manuscript is riddled with grammatical, spelling, and
            stylistic errors.",
        5: "Contains a number of minor errors that are slightly
            distracting.",
        10: "The writing is polished and professional, with no
            noticeable errors."
    },
    "figure_and_table_quality": {
        0: "Figures/tables are illegible, poorly designed, misleading
            , or incomprehensible.",
        5: "Figures/tables are understandable but could be better
            designed or labeled.",
        10: "Figures/tables are exceptionally well-designed,
            informative, and easy to interpret."
    },

    # VIII. Future Impact & Influence
    "aha_factor": {
        0: "The paper offers no new insights or 'aha' moments.",
        5: "The work is solid but does not provide any particularly
            surprising or deep insights.",
        10: "Provides a fundamental insight that changes how one
            thinks about the problem."
    },
    "foundation_for_future_work": {
        0: "The work is a dead-end and is unlikely to be built upon
            by others.",
        5: "The work may be cited but is unlikely to form the basis
            for much future research.",
        10: "The work is highly likely to be foundational, frequently
             cited, and built upon."
    },
    "interdisciplinary_appeal": {
        0: "The work is highly niche with no relevance outside its
            immediate sub-field.",
        5: "The work may have some relevance to adjacent fields, but
            this is not explored.",
        10: "The work has clear and significant potential to
            influence other research fields."
    },
    "scalability": {
        0: "The method is a 'toy' example that cannot scale to real-
            world problems.",
        5: "The method can scale to moderately sized problems but has
             clear limitations.",
        10: "The method is highly scalable and applicable to large,
            complex, real-world scenarios."
    },
    "simplicity_and_elegance": {
```

```
        0: "The core idea is overly complex, convoluted, or ad-hoc.",
        5: "The idea is functional but lacks elegance or simplicity
            .",
        10: "The core idea is surprisingly simple, elegant, and
            powerful."
    },
    "resource_provision": {
        0: "No code, data, or supplementary materials are provided.",
        5: "Resources are provided but are poorly documented,
            incomplete, or hard to use.",
        10: "Provides high-quality, well-documented, and easy-to-use
            open-source resources."
    },
    "educational_value": {
        0: "The paper is too confusing or poorly written to have any
            educational value.",
        5: "The paper could be used for teaching, but would require
            significant explanation.",
        10: "The paper is an excellent pedagogical tool for
            explaining a key concept."
    },
    "debate_provocation": {
        0: "The paper is unlikely to generate any discussion in the
            community.",
        5: "The paper might generate some minor discussion on
            specific points.",
        10: "The paper is highly likely to spark important new
            discussions or productive debates."
    },
    "practical_applicability": {
        0: "The work has no foreseeable practical application.",
        5: "The work has some potential for application, but it is
            distant or faces major hurdles.",
        10: "The work has clear, direct, and significant potential
            for real-world application."
    },
    "societal_impact": {
        0: "The work has no discernible societal impact.",
        5: "The work could have a minor or indirect societal impact
            .",
        10: "The work has the potential for major positive or
            negative societal consequences."
    },
    "memorable_takeaway": {
        0: "There is no clear or memorable take-home message.",
        5: "The take-home message is present but not particularly
            strong or clear.",
        10: "The paper has a clear, impactful, and memorable take-
            home message."
    },
    "efficiency_improvement": {
        0: "Offers no improvement in efficiency, cost, or performance
            ; may even be worse.",
        5: "Offers a marginal or incremental improvement over
            existing solutions.",
        10: "Offers a substantial, order-of-magnitude improvement in
            efficiency or performance."
    },
    "opens_new_directions": {
        0: "This work closes a line of inquiry rather than opening a
            new one.",
        5: "The work suggests minor variations on existing research
            lines.",
```

```
        10: "The work has the potential to open up entirely new and
            fruitful avenues of research."
    }
}
% \end{verbatim}
```

## F  PROMPTS

Here we provide the prompts used for each role in APRES. Including the prompt for reviewing a paper using a predefined rubric, the prompt for proposing new rubric, and the prompt for paper revision.

---

**Reviewer Prompt**

```
You are reviewing a paper for a top-tier, highly selective academic
    conference.
Please evaluate the following paper according to these criteria
   with their scoring guidelines:

**{rubric_item}**: {Rubric item description}.
  Score 0: {Score 0 guideline}
  Score 5: {Score 5 guideline}
  Score 10: {Score 10 guideline}

Given the highly selective nature of this top conference,
please apply rigorous standards in your evaluation.
Only exceptional work should receive scores of 8-10,
while work with significant flaws should receive lower scores.

Please provide your response in the following JSON format
wrapped in ```json ``` tags:
```json
{
  "{rubric_item}": {"score": <0-10>, "feedback": "<detailed feedback
      >"},
}
```
```

---

**Rubric Proposer Prompt**

```
You are an expert in research evaluation.
Your task is to propose a set of evaluation rubrics designed
to predict the future citation count of a research paper.

Your output must be two Python dictionaries:
`EVALUATION_RUBRIC` and `SCORING_GUIDELINES`.

1. `EVALUATION_RUBRIC`: Keys should be short, snake_case strings
(e.g., `novelty`), and values should be a clear question defining
   the criterion.

2. `SCORING_GUIDELINES`: Keys must match those in `
    EVALUATION_RUBRIC`.
Values should be a nested dictionary using a 0-5-10 scoring scale
(0=poor, 5=average, 10=exceptional) with a brief, clear description
    for each score.
```

```
The rubrics should focus on factors that make a paper influential
    and highly cited.

Example Format:
```python
EVALUATION_RUBRIC = {
    "clarity": "Is the paper's writing and structure exceptionally
        clear?",
}

SCORING_GUIDELINES = {
    "clarity": {
        0: "The paper is confusing or unintelligible.",
        5: "The paper is understandable but lacks precision.",
        10: "The paper is exceptionally clear and unambiguous."
    },
}
Please generate the complete EVALUATION_RUBRIC and
    SCORING_GUIDELINES dictionaries.
```

**Revision Prompt**

```
You are an expert academic editor.
Your goal is to rewrite the paper provided below to achieve a
    higher score based on
the given evaluation rubrics.

**Instructions:**
1. Work section-by-section (e.g., Abstract, Introduction, etc.).
2. For each section, first briefly list the key weaknesses you are
    fixing by referencing
the `rubric_item_key`.
3. Then, provide the improved, rewritten version of that section.
4. Focus only on improving the writing, framing, and structure.
**Do not change the core data, findings, or results.**

---
### **Context: Evaluation Rubrics**
{EVALUATION_RUBRIC}
{SCORING_GUIDELINES}

---
### **Context: Original Paper**
{Paper text}

---
```

## G   USER STUDY ANNOTATION GUIDELINES

**Annotation Guidelines: Evaluating AI-Revised Scientific Papers.**

**Context.** We are interested in the ability of LLMs to judge scientific paper quality and potentially improve papers. To this end, we will be annotating pairs of versions of a paper excerpt, and asking annotators to select which one they prefer. This feedback will help us understand and improve LLMs as scientific writing assistants.

**The Task in a Nutshell.** Annotators will be given two versions of a paper excerpt (e.g., abstract, introduction, or a specific section). The papers may be human- or LLM-generated. After reading

and comparing both, they will select their preference and provide a brief justification based on the evaluation criteria below.

**Core Evaluation Framework: NeurIPS Guidelines.** The evaluation should be grounded in the principles of a high-quality scientific paper, as defined by the NeurIPS reviewing guidelines. We are primarily concerned with the *quality of the writing and presentation*, as the core scientific contribution will not change between versions.

| Criterion | Key Questions |
|---|---|
| Clarity | Is the text easy to read and understand? Is the language precise and unambiguous? Does the text flow logically? Is the core message communicated effectively? |
| Correctness | Does one version contain apparent technical inaccuracies or oversimplifications (we guarantee that the contribution of both papers is the same). |
| Contribution & Impact | Does one version do a better job of framing the paper's contribution? Is the significance of the work made more apparent and compelling in one over the other? |
| Presentation & Style | Is the grammar, spelling, and punctuation flawless? Is the tone appropriately academic and professional? Is the text concise, or is it unnecessarily verbose? |

**Deliverables.**

1. For each pair, a preference (or "neutral" / no strong preference, which should be used very sparingly).
2. For each pair, a justification. This should briefly explain why one version was selected, referring to the evaluation criteria above (e.g., "I prefer this version because it replaces a long, convoluted sentence with two clearer ones, improving the logical flow.")

**Example Scenarios.** Here are a few hypothetical examples to guide your thinking.

**Example 1:**

- **Version A:** "Our proposed methodology, which leverages a novel attention mechanism within a transformer architecture, is demonstrated through extensive experimentation to achieve a marked improvement in performance, surpassing existing state-of-the-art models on the benchmark dataset."
- **Version B:** "We propose a new attention mechanism for transformer architectures. Extensive experiments show our method achieves state-of-the-art performance on the benchmark dataset."
- **Decision: Prefer Version B.**
- **Justification:** "The B version is far more direct and concise. It removes unnecessary jargon like 'demonstrated through extensive experimentation' and and 'marked improvement,' making the core contribution easier to grasp immediately."

**Example 2:**

- **Version A:** "While our model performs well on datasets with balanced class distributions, its accuracy degrades non-linearly on highly skewed distributions, a key limitation for real-world deployment."
- **Version B:** "Our model has issues with certain datasets. The accuracy is lower on skewed distributions, which is a limitation."
- **Decision: Prefer Version A.**
- **Justification:** "Version B oversimplified the text to the point of losing critical information. The original's use of 'non-linearly' and 'highly skewed' are precise, important technical details that the revision completely omits. Version A is better because it is more correct."

**Example 3: Neutral / No Strong Preference**

- **Version A:** "In this section, we will describe the dataset used for our experiments. Furthermore, we will detail the preprocessing steps that were applied."
- **Version B:** "This section describes the dataset used for our experiments and details the applied preprocessing steps."

- **Decision: Neutral / No Strong Preference.**
- **Justification:** "Both versions are perfectly acceptable."

## H    EXAMPLE PAPERS

We provide a few example papers before and after the revision by APRES. They can be found in the same zip file with this supplementary material. Qualitatively, APRES tends to revise a paper with systematical focus on enhancing its communicative power, guided by the high-signal items in the discovered rubric. The revision strategy prioritizes sharpening contribution framing, primarily by strengthening the explicit "gap identification" and "contribution articulation" in the abstract and introduction to make the paper's novelty immediately apparent. Concurrently, the agent works on optimizing logical structure, refining section transitions and restructuring arguments to improve the overall narrative flow. Furthermore, revisions systematically enhance **clarity and conciseness**, eliminating unnecessary jargon and simplifying convoluted syntax to maximize readability. These presentation-focused edits lead to the high human preference scores observed in our evaluations.

## I    EFFECTIVENESS OF LLM REVIEWERS

In this section, we provide an additional study on how effective LLM review systems are from the perspective of how well the LLM reviews correlates with the actual conference outcomes and how consistent the LLM review systems are with respect to the consistency of human reviewers.

### I.1    GLICKO2 RATING

The first step is to study whether or not LLM review systems can rank papers that correlate with how actual conference would. To measure this, we designed a rating system based on the Glicko2 scores (Glickman, 2012). Glicko2 ratings are computed in a pairwise fashion, specifically, we first generate a review for each paper in our test dataset using the discovered rubric from APRES. Then we randomly sample paper pairs and their reviews and ask an LLM judge to compare the two papers to determine which paper is 'better'. We continue the pairwise paper comparison until the overall rating of all paper stabilizes, or the total number of comparison runs out. We set the total number of pairwise comparisons to 20,000. The LLM judge prompt is provided below.

```
Judge Prompt

You are an expert reviewer comparing two research papers. You have
    access to the full text of both papers and their detailed
    reviews.
The reviews are generated by the following system prompt:
{review_instruction_form}

Below are the reviews of the two papers:

Paper A:
Title: {paper_a_id}
Text: {paper_a_text}
Reviews: {paper_a_reviews}

Paper B:
Title: {paper_b_id}
Text: {paper_b_text}
Reviews: {paper_b_reviews}

Based on the papers and their reviews, which paper is better
    overall? Consider all aspects including:
- Technical soundness and correctness
- Novelty and originality
- Significance and impact
```

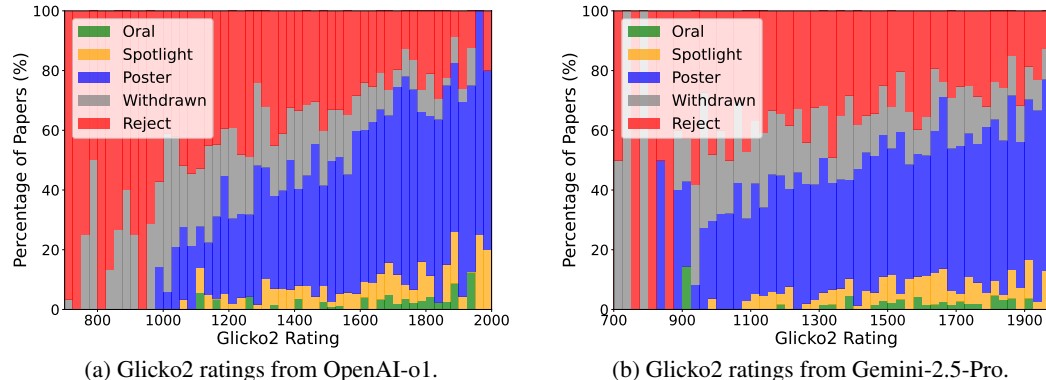

(a) Glicko2 ratings from OpenAI-o1.   (b) Glicko2 ratings from Gemini-2.5-Pro.

Figure A4: LLM-derived Glicko2 ratings strongly correlate with final conference decisions. As the Glicko2 rating increases, the proportion of rejected papers (red) decreases while the proportion of accepted papers—posters (blue), spotlights (yellow), and orals (green) steadily rises.

```
- Clarity of presentation
- Quality of experiments and evaluation
- Overall contribution to the field

Respond in the following format:

THOUGHT:
<Your reasoning for the comparison>

DECISION:
```json
{{
    "confidence": 1-5 (1=very uncertain, 5=very confident),
    "reasoning": "Brief explanation of why one paper is better",
    "score_difference": 1-10 (how much better the winner is, 1=
        slightly better, 10=much better)
    "winner": "A" or "B",
}}
```
```

**Results and Analysis.** This Glicko2 rating system can create a continuous quality ranking for all papers. We validate this ranking against the official conference outcomes. The results, visualized in Fig. A4, demonstrate a clear and strong positive correlation between a paper's Glicko2 rating, as determined by our LLM judge, and the final conference decision. As the Glicko2 rating increases along the x-axis, the composition of paper types changes dramatically. At lower rating levels, the vast majority of papers are those that were ultimately rejected (red) by the conference. As the rating surpasses a certain threshold (around 1200), the proportion of accepted poster presentations (blue) begins to grow significantly. Most importantly, at the highest rating levels, the most prestigious outcomes, spotlight (yellow) and oral (green) presentations, emerge and constitute a substantial fraction of the papers. This trend confirms that the relative quality judgments made by the LLM in pairwise comparisons successfully capture the same signals that human committees use to distinguish exceptional work. This result validates LLMs with APRES discovered rubric can be used for constructing systems that provides ratings for papers that correlates well with human committees.

## I.2 REVIEWER CONSISTENCY

**Motivation.** A fundamental prerequisite for a fair and effective peer review system is consistency. If the outcome of a review is heavily dependent on which specific set of reviewers a paper is assigned to, the process can be considered arbitrary. Landmark studies, such as the NeurIPS consistency experiments in 2014 (Cortes & Lawrence, 2021) and 2021 (Beygelzimer et al., 2023), have quantita-

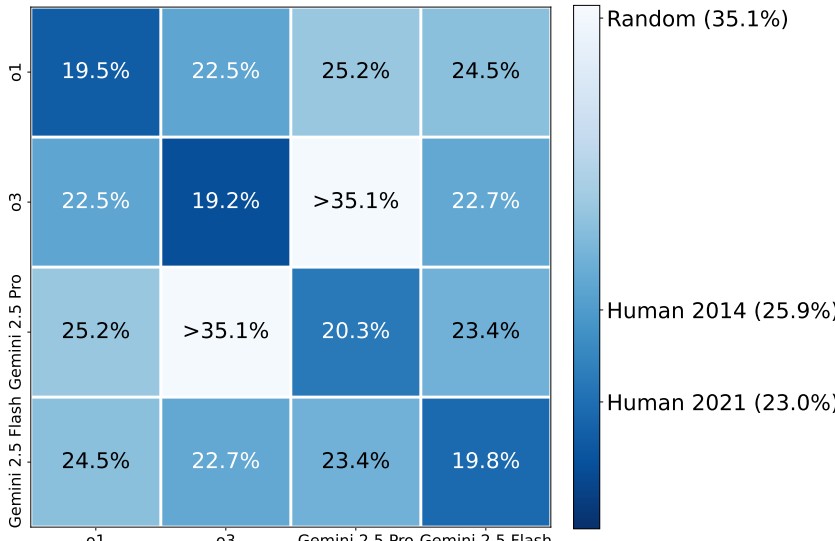

Figure A5: LLM-based review committees are more consistent than human committees. The heatmap shows the pairwise disagreement rate between various committee configurations. Each axis corresponds to a different base LLM judge (i.e., `OpenAI-o1`, `OpenAI-o3`, `Gemini 2.5 Pro`, and `Gemini 2.5 Flash`).

tively demonstrated the scale of this randomness in human peer review. In the 2021 experiment, two independent review committees disagreed on the final accept/reject decision for 23% of the papers they both evaluated (Beygelzimer et al., 2023). This led to the striking conclusion that if the review process were rerun, approximately half of the accepted papers would have been different. This high level of stochasticity not only impacts authors' careers but also raises questions about the reliability of the scientific record. Motivated by this challenge, we are interested in understanding whether or not LLM reviewers can offer a stable and less random evaluation process, therefore providing a more consistent and reliable signal for paper impact prediction as well as paper revision.

**Methodology.** To measure the consistency of our LLM-based evaluation, we designed an experiment to parallel the NeurIPS studies. We adopted a two-step process to arrive at a final decision, first establishing a continuous quality score for each paper before converting it to a binary outcome. We use the Glicko2 rating system from Sec. I.1 to obtain the ratings for each paper. Then, a binary decision, $d_p \in \{\text{Accept}, \text{Reject}\}$, is derived by thresholding these ratings. Papers with a Glicko2 rating in the top 25th percentile are labeled 'Accept', while the remaining 75% are labeled 'Reject'. This indirect approach was chosen over directly prompting an LLM for an accept/reject vote for several key reasons. First, it allows us to precisely control the acceptance rate (e.g., to 25%), mirroring the selectivity of top-tier conferences. Second, it mitigates the potential randomness and inherent biases an LLM might have in making a direct classification of accept/reject decisions. Third, it simplifies the task for the LLM judge, which only needs to perform a relative comparison between two papers with their reviews already provided, rather than making a global judgment of absolute quality.

To conduct a comprehensive consistency analysis, we measure the disagreement between any two distinct models, $i$ and $j$. For each model, we produce a complete set of binary decisions, $\{d_{p,i}\}_{p \in \mathcal{P}}$ and $\{d_{p,j}\}_{p \in \mathcal{P}}$. The Disagreement Rate ($DR$) between them is calculated as:

$$DR(i,j) = \frac{1}{|\mathcal{P}|} \sum_{p \in \mathcal{P}} \mathbb{I}(d_{p,i} \neq d_{p,j}),$$

where $\mathbb{I}(\cdot)$ is the indicator function. This enables the construction of a full consistency matrix, comparing every model against every other and against the human benchmarks of 23% (NeurIPS 2021) and 25.9% (NeurIPS 2014) disagreement. For the comparison with the same model, we rerun the same model twice to obtain two set of decisions for the papers.

**Results and Analysis.** The results of our consistency analysis are presented in Fig. A5, which shows a matrix of pairwise disagreement rates between four models, including `o1`, `o3`, `Gemini 2.5 Flash`, and `Gemini 2.5 Pro`. The primary finding is a strong and clear one: LLM-based evaluation can be significantly more consistent than human peer review. The high level of consistency is particularly pronounced within the same model family, as seen in the lower disagreement rate between `o1` and `o3`, as well as between `Gemini 2.5 Flash` and `Gemini 2.5 Pro`. This demonstrates that for a given base model, the final decisions are remarkably stable when using APRES discovered rubric and the Glicko2 ratings. While inter-model family agreement shows slightly more variability and in some cases, worse than the agreement of a random baseline, many inter-model family comparisons still outperform the human baselines. Overall, the evidence strongly suggests that the LLM-driven pipeline provides a more reliable and less random signal of paper quality compared to the established stochasticity of the human review process. This results established that LLMs can provide a stable signal, therefore validating APRES is reliable for providing meaningful revisions to research papers.

## J  RESOURCE COST ANALYSIS

We provide a comprehensive analysis of the computational and monetary footprint of the APRES framework. The estimates below are based on the observed token consumption for key LLM operations and utilize the per-million token pricing of advanced models (e.g., OpenAI o3 and Gemini 2.5 Pro) at the time of writing. The average paper consumed approximately $17-18k$ tokens after SciPDF parsing.

### J.1  STAGE 1: RUBRIC DISCOVERY (ONE-TIME COST)

This stage involves the iterative search process over $N = 26,707$ papers to discover the optimal rubric, run for $I_{RS}$ (200) iterations. A single scoring API call by the 'Reviewer' agent consumes approximately **6,000** input tokens and **2,000** output tokens per paper.

**Cost per Scoring API Call** ($C_{Score}$): We calculate the monetary cost for scoring a single paper based on observed token usage:

- o3: $\approx \$0.0056$ per paper
- Gemini 2.5 Pro: $\approx \$0.0275$ per paper

**Total Rubric Discovery Cost** ($C_{Total,RS}$): The total cost for the Rubric Search is calculated as a function of the number of search iterations ($I_{RS}$), the number of papers ($N$), and the cost per scoring call ($C_{Score}$):

$$C_{Total,RS} = I_{RS} \times N \times C_{Score}$$

We note that the MAE convergence curve (Figure 2) suggests the system achieves nearly complete performance convergence within the first **50** iterations, which significantly reduces the practical discovery cost:

- o3 Estimate (50 iterations): $\approx \$\mathbf{7,500}$
- Gemini 2.5 Pro Estimate (50 iterations): $\approx \$\mathbf{36,700}$

### J.2  STAGE 2: PAPER REVISION (PER-PAPER COST)

This stage runs the iterative Reviewer $\rightarrow$ Rewriter loop on a single paper for $I_{PR}$ (**120**) iterations. A single revision iteration consumes **6,000** input tokens and **1,500** output tokens. The total token consumption per paper over 120 iterations is **0.9** Million tokens.

**Cost per Paper Revision Loop** ($C_{Total,PR}$): The total cost for revising a single paper is calculated as a function of the number of iterations ($I_{PR}$) and the cost per revision iteration ($C_{Revision\ Iteration}$):

$$C_{Total,PR} = I_{PR} \times C_{Revision\ Iteration}$$

Based on the pricing of per 1M tokens of the models, the cost for the models per paper is:

- o3 Estimate: $\approx \$\mathbf{0.58}$ per paper
- Gemini 2.5 Pro Estimate: $\approx \$\mathbf{2.70}$ per paper

The resulting cost per paper revision is minimal relative to the potential value added to the manuscript, justifying our iterative, agentic approach.

