# OpenReview forum: "APRES: An Agentic Paper Revision and Evaluation System"
_ICLR.cc/2026/Conference — Submitted to ICLR 2026_

### Official Review · Reviewer_Dwr5 · 2025-10-19

**Soundness:** 3
**Presentation:** 2
**Contribution:** 3
**Rating:** 6
**Confidence:** 4

**Summary:**

This paper introduces **APRES**, an agent-based system that leverages LLMs to both **discover review rubrics predictive of future citation impact** and **revise scientific papers accordingly**. The system first performs an iterative search to learn a rubric that best predicts future citations using negative binomial regression. This learned rubric is then used to guide an LLM-based revision process aimed at enhancing paper clarity and presentation while preserving scientific content. Experiments on ICLR and NeurIPS papers show that APRES improves citation prediction accuracy by 19.6% over baseline methods, and human evaluators generally prefer the revised papers. Overall, APRES aims to complement human peer review by offering consistent, impact-oriented feedback and revisions.

**Strengths:**

* **Novel Framework:** The paper presents a novel approach that connects automated paper evaluation with revision based on citation prediction. Viewing citation count as a proxy for paper quality is a reasonable, although indirect, assumption, and optimizing revisions with respect to expected impact is an interesting and meaningful direction.
* **Practicality and Transparency:** The inclusion of concrete examples of papers revised by APRES is commendable and increases transparency.
* **Empirical Usefulness:** Experimental results suggest that APRES improves submission quality, both in terms of citation prediction accuracy and human preference for the revised versions.

**Weaknesses:**

* **(Major) Lack of Cost Analysis:** My major concern is the absence of any analysis of the computational or monetary cost of APRES. Given the high pricing of advanced models like o1 and o3, it is essential to quantify the overall cost of rubric discovery across the dataset and the cost per paper revision. I would be willing to adjust my evaluation if such details were provided.
* **(Minor) Scope of Framework:** It is a bit unclear if APRES can only be applied to AI research papers or also applied to other areas.
* **(Minor) Clarity of Framework Presentation:** Figure 1 is difficult to interpret on its own. On first reading, it appears as though the rubric discovery and paper revision processes happen simultaneously, which is not the case. Although both rely on a search scaffold, revising the figure to more explicitly separate and explain these stages would greatly improve clarity.
* **(Minor) Missing Related Work:** The paper does not reference some relevant recent work, including:
  * [*ReviewScore: Misinformed Peer Review Detection with Large Language Models*](https://arxiv.org/abs/2509.21679)
  * [*Position Paper: How Should We Responsibly Adopt LLMs in the Peer Review Process?*](https://openreview.net/forum?id=KZ3NspcpLN)

**Questions:**

* Did the authors use the APRES framework to revise this manuscript?
* Could you provide more details on the recruitment process and expertise of the human evaluators?
* Lines 236–238: This sentence is unclear and should be revised for clarity.

**Details Of Ethics Concerns:**

Even though there are some unclear parts regarding the recruitment of human evaluators, I decided not to flag this as ethical concern.

---

> ### Author Response · Authors · 2025-11-22
>
> We thank **Dwr5** for their thoughtful and detailed assessment. We are delighted that they recognize the novelty of our framework, particularly the approach of connecting automated paper evaluation directly with revision guided by predicted citation impact, confirming this as an interesting and meaningful direction for research. We are also grateful that they commend the practicality and transparency of our work, specifically noting the inclusion of concrete revision examples, and acknowledge the empirical usefulness demonstrated by the improvement in citation prediction accuracy and the strong human preference for the revised papers.
>
> > Dwr5-1: Cost Analysis
>
> A1: We will add a comprehensive Resource Cost Analysis to the Appendix.
> The average paper in our dataset consumes approximately $6,000$ tokens after SciPDF parsing. We detail costs below for the two stages: the one-time Rubric Discovery (Stage 1) and the Revision Loop (Stage 2). Importantly, the costs below are based on current fees. These costs could decrease in future as more performant models are made available at lower costs.
>
> ### Stage 1: Rubric Discovery (One-Time Cost)
>
> This stage involves scoring the entire dataset of papers ($N = 26,707$) iteratively. A single scoring API call consumes approximately $\mathbf{6,000}$ input tokens and $\mathbf{2,000}$ output tokens per paper.
> Cost per Scoring API Call ($C_{Score}$):
> We calculate the cost based on the input and output tokens consumed for each scoring operation:
> - o3: $\approx 0.0056$ dollar per paper
> - Gemini 2.5 Pro: $\approx 0.0275$  dollar per paper
>
> Total Rubric Discovery Cost ($C_{Total, RS}$):
> The total cost for the full Rubric Search is calculated as:
> $$C_{Total, RS} = I_{RS} \times N \times C_{Score}$$
> We note that analysis of the MAE curve (Figure 2) suggests the system achieves nearly complete performance convergence within the first 50 iterations, which significantly reduces the practical cost of discovery:
> - o3 Estimate (50 iterations): $\approx \mathbf{\$7,500}$
> - Gemini 2.5 Pro Estimate (50 iterations): $\approx \mathbf{\$36,700}$
>
> ### Stage 2: Paper Revision (Per-Paper Cost)
>
> This stage runs the iterative Reviewer $\rightarrow$ Rewriter loop on a single paper for $I_{PR}$ ($\mathbf{120}$) iterations. A single revision iteration consumes $\mathbf{6,000}$ input tokens and $\mathbf{1,500}$ output tokens. The total token consumption per paper is $\mathbf{5.4 \text{ Million tokens}}$.
> The total cost for revising a single paper is calculated as a function of the number of iterations:
> $$C_{Total, PR} = I_{PR} \times C_{Revision\ Iteration}$$
> Based on the pricing of per 1M tokens of the models, the cost for the models per paper is:
>
> - o3 Estimate: $\approx \mathbf{\$0.58}$ per paper
> - Gemini 2.5 Pro Estimate: $\approx \mathbf{\$2.70}$ per paper
>
> The cost per paper revision is minimal relative to the potential value added to the manuscript, justifying the iterative agentic approach.
>
>
>
> > Dwr5-2: It is unclear if APRES can only be applied to AI research papers or other areas.
>
> A2: This is a really interesting question. Our experiments used AI conference papers (ICLR/NeurIPS) due to data availability. However, the methodology (Rubric Search) is domain-agnostic. If applied to a Biology corpus with citation data, the system would simply discover a different rubric (e.g., prioritizing "Experimental Reproducibility") relevant to that field. We will update the discussion to reflect this observation.
>
> > Dwr5-3: Figure 1 clarity. Figure 1 implies simultaneous processes rather than sequential stages for rubric discovery and paper revision.
>
> A3: We will redesign Figure 1 to explicitly show the two stages as sequential blocks, clarifying that the Rubric Search happens before the Paper Revision. Thanks for the suggestion.
>
> > Dwr5-4: Add citations for ReviewScore and Position Paper.
>
> A4: The ReviewScore paper first appeared on arXiv on September 25th 2025, just days before the ICLR deadline. It focuses on detecting misinformed reviews which could be supplementary to our work which focuses on discovering useful rubrics that can be used to revise a paper for better potential impact. The position paper argues against the use of LLMs to generate review texts end-to-end and advocates for using LLMs for checking reproducibility, relevance of citations, and assisting with ethic review. We will cite and discuss these two papers.

---

> > ### Author Response · Authors · 2025-11-22
> >
> > > Dwr5-5: Did the authors use the APRES framework to revise this manuscript?
> >
> > A5: As stated in “Appendix A: LLM Use”, we did not use APRES to write or revise this manuscript. We believe it is important to maintain a clear separation between the tool and the evaluation of the tool during the blind review process to avoid any potential ethical ambiguity.
> >
> > > Dwr5-6: Details on human evaluator recruitment.
> >
> > A6: Our human evaluators were recruited via specialized, compliant public online platforms (e.g., Outlier, Turing). These platforms ensured that all annotators met the necessary professional expertise, specifically requiring them to be PhD-level researchers in machine learning (or actively pursuing one). Utilizing these platforms confirms that all annotators met the stated conditions and that the process conforms to relevant regulations for human annotation studies, ensuring the integrity and high standards of our pairwise preference evaluation.

---

> > > ### Comment · Reviewer_Dwr5 · 2025-11-23
> > >
> > > I think the authors well recognized my concerns and raised valid points to relieve them. However, as it is possible to upload a revised version of the paper, I would recommend that the authors to do so.

---

> > > > ### Author Response · Authors · 2025-11-28
> > > >
> > > > We thank reviewer Dwr5 for the thoughtful feedback and are glad that our earlier response addressed the main concerns. In line with the suggestions, we have made the following revisions to the paper:
> > > >
> > > > 1. **Related Work**: We have updated the related work section to include the two cited works mentioned in the review and clarified their relation to our contributions.
> > > > 2. **Cost Analysis**: We have added a dedicated cost analysis section in the appendix to provide transparency regarding computational and monetary costs.
> > > > 3. **Figure Clarity**: The original Figure 1 has been split into Figure 1 and Figure 2 to clearly separate the two stages of APRES. The captions now explicitly describe the inputs, outputs, and logic of each node in the search process.
> > > > 4. **Broader Applicability**: Section 5 now includes additional discussion on how APRES can generalize to other scientific domains beyond machine learning.
> > > > 5. **Human Evaluator Details**: We have updated the ethics statement with more details on the human evaluator recruitment.
> > > > 6. **Content Preservation**: We have added a new appendix C in the appendix to provide a new analysis we have performed which shows that our approach does not change results in Tables or Figs.
> > > >
> > > > We appreciate the reviewer’s constructive input and believe these updates have strengthened the clarity and completeness of the paper.

---

### Official Review · Reviewer_7q2X · 2025-10-26

**Soundness:** 2
**Presentation:** 1
**Contribution:** 2
**Rating:** 2
**Confidence:** 4

**Summary:**

The paper proposes a method called APRES, which is powered by LLMs. It updates a scientific paper based on an evaluation rubric that is highly predictive of future citation counts. The paper demonstrates APRES improves future citation and that its revision process yields papers preferred by human evaluators. The paper seeks to augment human expert reviewers, using LLMs to help authors review and revise manuscripts before submission.

**Strengths:**

1. The idea of using LLMs to assist paper writing is promising.
2. Agentic paper revision and evaluation is intuitive to help researchers in writing papers.

**Weaknesses:**

1. The core design of this paper uses the number of **citations** as a metric to improve given papers' presentation and readability, which is questionable. The authors cite Ante (2022) as support. However: a) The claim of Ante's paper is "Our analysis furthermore shows that higher readability scores significantly relates to the likelihood of articles not receiving any citations", which **contradicts the claim of this paper**. b) Even if readability (x) can increase or decrease scientific impact (y), this is an x→y relationship. We cannot reverse it to infer y→x, which constitutes a **logical flaw**.
2. The authors claim that the **challenge** of LLMs for review and revision lies in "inadvertently modifying scientific claims or deviating from accepted academic styles". However, the design of their method targeted at increasing presentation and readability does not seem to specifically address this challenge. The authors claim that their method can "preserve its core scientific content", but there is no targeted design for this purpose.
3. The **presentation** of this paper needs a significant improvement. Currently, the method description is unclear. For example, Figure 1 is highly unclear, it is hard to understand the pipeline. The description in the Method section and its connection to this illustrative figure are also unclear.

**References**

[1] Lennart Ante. The relationship between readability and scientific impact: Evidence from emerging technology discourses. Journal of Informetrics, 16(1):101252, 2022.

**Questions:**

1. What do the nodes in Figure 1 represent? Why is the structure designed this way? Is there any basis for such a design? And is there an ablation study to support this design?
2. An interesting point is: Was this paper assisted by the system it claimed? If not, why not? If yes, then it seems the system is not helpful for paper writing, as the readability of this paper appears to be poor.

---

> ### Author Response · Authors · 2025-11-22
>
> > 7q2x-1: Using number of citations as a metric to improve a paper.
>
> A1: As discussed on L168-172 and L437-441, we fully agree that citation count is an imperfect measure of a paper’s quality. However, they remain one of the most widely used and available proxies for scientific influence.
>
> (a) Thank you for pointing out our imprecise citing of Ante (2022). Ante shows that lexical complexity in abstracts correlates positively with citations, especially in mature fields, and does not argue against clarity itself—only that overly simple, low-complexity abstracts tend to receive fewer citations.
> Our system is not designed to optimize for surface-level simplicity. In fact, the automatically discovered rubric consistently emphasizes clarity of contribution, logical structure, and field relevance, none of which correspond to the lexical simplification measures examined by Ante. The empirical evaluation further supports this: human experts preferred APRES-generated revisions in 79% of comparisons, indicating that the system improves the substantive clarity and presentation of ideas rather than producing the kinds of superficial simplifications that Ante finds negatively associated with impact.
> We will update the manuscript to more accurately characterize  their findings (Ante 2022)  in the context of our work.
>
> (b) We will revise our paper to make it clearer that APRES is only designed to revise the papers to maximize for a predicted impact of the paper and not the actual future impact. However, we still believe that clear communication of scientific information is the key to achieve higher impact and that APRES can help authors improve their papers towards this goal.
>
> > 7q2x-2: The method lacks specific design elements to guarantee the preservation of scientific claims beyond general prompts.
>
> A2: We employ more than just prompts. We utilize a deterministic code-based filter to prevent edits to tables and figures as mentioned in Line 723. This guardrail directly utilizes the SciPDF parser’s design for classifying parsed texts from the pdf into categories of ‘section’, ‘figures’, and ‘tables’. Please note that SciPDF is a deterministic parser that directly uses the PDF metadata for parsing, therefore the process is accurate. After the parsing, we lock all text contents classified as tables, and as a result they cannot be updated/changed during the revision process. We have rerun 500 papers for validating if the scientific details of the papers are changed and manually checked all proposed changes. In all of the cases examined, none of the results is changed by our system due to the fact that our guardrail is able to block any proposed changes to table contents. We agree that this is a very important step, and we will update the main paper text to better describe the guardrails we impose and to describe the results of our validation experiment. Thanks for pointing this out.
>
>
> > 7q2x-3: The method description and figure are unclear.
>
> A3: We will revise the caption and labels in Figure 1 to improve clarity. The nodes represent States in the search process. In Stage 1, a Node is a candidate Rubric. In Stage 2, a Node is a candidate Revision of the paper text. The edges represent the agent's actions (modifying the rubric or rewriting a section). This representation of the search process is common in the domain of agent frameworks, e.g., AIDE (Jiang et al 2025), MultiAIDE (Zhao et al. 2025a), and Prompt breeder (Fernando et al., 2024), use similar notation. We will use the extra page provided in the camera ready version to clarify the description of the method and more clearly link it to the steps outlined in Figure 1.
>
> > 7q2x-4: Was this paper itself written/revised by the system?
>
> A4: As stated in “Appendix A: LLM Use”, we did not use APRES to write or revise this manuscript. We believe it is important to maintain a clear separation between the tool and the evaluation of the tool during the blind review process to avoid any potential ethical ambiguity.

---

> > ### Comment · Reviewer_7q2X · 2025-11-24
> >
> > I think the authors are avoiding the key issues and have not addressed my concerns.
> >
> > 1. I agree that "citation count remains one of the most widely used and available proxies for scientific influence." However, my question concerns **the rationality of using citations to improve paper presentation**.
> >
> >     1.1. The original statement in Ante (2022) I mentioned in my previous comments **contradicts** the claims in the paper. The paper **incorrectly cites** this work as their core motivation, and the authors’ rebuttal is not convicing.
> >
> >     1.2. Citation count is jointly influenced by **multiple factors** such as writing, contributions, and novelty. Yet the paper inversely uses citation count to improve writing. This constitutes a reversal of cause and effect, and it is statistically unreasonable.
> >
> > 2. I do not consider the use of SciPDF parser a domain-specific innovation. It is more of an **engineering implementation**. I maintain my view that this paper does not provide a **targeted and innovative solution to the proposed challenge**, which undermines the contribution of this work.
> >
> > 3. I reserve my opinions on the paper’s writing and presentation. Taking visualization as an example, while many papers do use tree structures to illustrate multi-step reasoning processes, this is not my concern. Figure 1 attempts to present the inference of two stages within a single node. However, the **node’s inputs and outputs, data flow, stage transitions, and node selection** are all unclear.
> >
> > In summary, I maintain my score as the authors do not address my concerns.

---

> > > ### Author Response · Authors · 2025-11-28
> > >
> > > We respectfully note that the reviewer’s concern appears to stem from a misunderstanding of what this paper claims as its core contribution. Engineering components such as the use of SciPDF are implementation choices and are not positioned as contributions of this work.
> > > To clarify, the main contributions of this paper are:
> > >
> > > 1. **A rubric discovery method** that identifies evaluation criteria whose scores are strongly aligned with actual citation counts, enabling a data-driven measure of impact rather than relying on fixed, human-defined rubrics.
> > > 2. **An automated paper revision system** that leverages this discovered rubric to guide iterative revisions, and demonstrably produces revised papers that are preferred by human expert judges.
> > >
> > >
> > > These contributions address the central challenge we focus on: how to provide actionable, evaluation-aligned revision signals that improve clarity and presentation. The engineering details are supportive infrastructure, not the novelty claim, and do not affect the substantive contribution of the work.
> > >
> > >
> > > > **7q2X-Followup-1.1**: Ante 2022 contradicts the claims in the paper
> > >
> > > A1.1: We did not cite Ante 2022 as the main motivation for our work. In fact, we have only cited Ante 2022 once in the paper to state that the readability of a paper has been shown to be correlated with its scientific impact. Other cited works, such as Ryba etal 2021, show that more accessible writing styles can improve the readers confidence. We have revised the paper to more accurately reflect the work by Ante 2022.
> > >
> > >
> > > > **7q2X-Followup-1.2**: Citation count cannot be used to improve writing
> > >
> > > A1.2: We believe the reviewer mixes up *causal inference* with *predictive/statistical use of a proxy*.
> > > 1. Multiple factors do not equal to invalid proxy. Even though the predicted citation count is a signal about multiple causes, it can still be used as a noise proxy for one of its causes. This is often used in many scenarios:
> > >   - Predicted citation counts as a proxy for writing quality/clarity (even though other factors like novelty also matters)
> > >   - Likes/views as a proxy for content engagement (even though timing and platform algorithms matter).
> > > 2. Using predicted citations as selection metric doesn’t assume or invert causality
> > >   - Employing predicted citation count as a scoring signal in a search procedure **does not claim that citations cause good writing**, nor that increasing citations will mechanically improve the text. The method simply treats predicted citations as a **correlated proxy** for desirable properties such as clarity, relevance, contribution framing. These properties tend to co-occur with higher impact papers. Leveraging a correlated outcome as an optimization signal is a predictive use of association, not a causal claim. Therefore, it does not constitute a “reversal of cause and effect”; it only assumes that citation predictions contain **some statistical signal** about impactful presentation, even if that signal is noisy and influenced by other factors.
> > >   - Moreover, we validated this assumption empirically. We conducted a **human evaluation** with PhD-level researchers who rated the revised papers on clarity, coherence, and overall presentation similar to the NeurIPS review guidelines. Their assessments indicated consistent improvements over the original versions. This provides independent support that the system enhances writing quality, rather than merely optimizing for a numerical proxy.
> > >
> > > Additionally, we acknowledge that citation counts is an imperfect metric for evaluating a paper as stated in Line 187 and 188. However, as shown in the paper, it is a useful proxy. Our method is also agnostic to the precise metric used, as long as it provides some signal wrt paper quality.
> > >
> > >
> > > > 7q2X-Followup-2: The use of SciPDF is only an engineering implementation
> > >
> > > A2: We agree that the use of SciPDF Parser is an engineering choice rather than a conceptual contribution. Importantly, we do not position SciPDF or any parsing component as a novelty claim in this work. Our contributions are independent of that implementation detail and remain unchanged.
> > > This paper does not frame “maintaining scientific content during rewriting” as its primary research challenge. The core problem we address is how to generate **useful, actionable revision feedback** that helps authors improve clarity, structure, and presentation. Our method APRES proposes a novel review-and-revision search loop and demonstrates, through human evaluation with experienced researchers, that the resulting revisions are consistently judged to be clearer and better presented. This is the contribution we intend to make, and it is not contingent on the choice of parsing tool.
> > > In the revised paper Appendix C, we have included a new analysis we performed which shows that our approach does not change results in Tables or Figs.

---

> > > > ### Author Response · Authors · 2025-11-28
> > > >
> > > > > 7q2X-Followup-3: Clarity of Figure 1
> > > >
> > > > A3: We appreciate the reviewer’s concern regarding the clarity of Figure 1. We have updated the text and figures to improve clarity in light of the suggestions. We are also happy to make additional changes if there are any specific sections that are unclear. In the revised version, we have separated the original figure into **Figure 1** and **Figure 2** to explicitly illustrate the two distinct stages of APRES: rubric discovery and paper revision. We have also updated the captions to clearly describe the inputs, outputs, and internal logic of each node within the search process. We hope this restructuring makes the workflow more transparent and easier to follow.

---

### Official Review · Reviewer_381o · 2025-10-30

**Soundness:** 2
**Presentation:** 3
**Contribution:** 3
**Rating:** 4
**Confidence:** 3

**Summary:**

This paper presents an end-to-end system for paper revision called APRES. APRES has two components. 1) The first component is a routine that discovers a paper evaluation rubric that is highly predictive of future citation counts. This component leverages two agents, a proposer and scorer, to propose rubrics that are then evaluated downstream as features to a negative binomial regression model. 2) The second component is a paper improvement routine that, given the discovered rubric, iteratively reviews and revises a given paper to score higher on the rubric.

Evaluations/Results:

The paper evaluates the first component by comparing to baselines along how well they can predict actual citation counts (scored by MAE). Results show that the iterative search procedure leads to a lower MAE than baselines.

The paper then evaluates the second component by showing that iterating on the discovered rubric (as opposed to rubrics discovered via baselines for component 1) leads to higher scores along the discovered rubric. Lastly, they show that human researchers prefer the papers revised with APRES to original papers in a pairwise task.

**Strengths:**

1. Problem impact/significance: The paper presents an end-to-end system for the impactful and timely problem of using LLMs to generate useful feedback for paper revision.

2. The idea of using an LLM-based agent to iteratively generate rubrics that are predictive of citation count based on text content appears to be novel, and the proposed MultiAIDE search method outperforms existing methods for predicting future citation count.

3. The human evaluation results scoring paper quality suggest that APRES can successfully improve paper quality.

**Weaknesses:**

1. As noted in the abstract, it is important that any paper revision system not revise core scientific content in a paper. While there are some constraints placed on the system to limit revision of such content, none of the evaluations of APRES and revised papers measure whether scientific content changed. This to me is one of the key weaknesses of this work, as a paper revision system that always changed a lot of scientific content such that the results looked more positive would be expected to have much higher potential impact, but such a revision system would be broken.
2. Another weakness in the evaluation is that one of the main evaluations of the paper revision component uses impact scores from the discovered rubric in the first component as ground truth signal for impact (see Figure 3). However, this leads to an unfair comparison with baselines, as the discovered rubric method is directly optimizing against the rubric used in the eval and thus hillclimbing on it, but the other methods are not. It is thus not surprising that discovered rubric leads to the best results. This evaluation would be strengthened if somehow another metric for paper improvement were used at evaluation time, such as human ratings of paper improvement.

3. There are some places in the paper that could benefit from increased clarity:
    - How is PromptBreeder different than the proposed method? Would be helpful to have this written out.
    - It seems that "MultiAIDE" refers to the iterative approaches in both the rubric discovery and paper revision components. It would help clarify the paper to separately name each of these components.
    - See other minor points below in "Questions"

**Questions:**

Minor notes:
- Line 235: "the text of the paper, it is prompt" -> "the text of the paper, and it is prompted..."
- Line 336: "constraint" -> "constrained"
- Line 234: Term "MultiAIDE" is used as if it has been introduced earlier in the paper, but it hasn't (only AIDE has been)

Questions:
- One way of improving papers would be to directly optimize for future paper impact rather than an intermediate rubric, ie simply using predicted # of citations as the scoring function in the second component. Why did authors not try such a method?

---

> ### Author Response · Authors · 2025-11-22
>
> We thank **381o** for their review and for recognizing the impact and timeliness of our end-to-end system for automated paper revision. We are encouraged that they find the idea of iteratively discovering citation-predictive rubrics to be novel and that they appreciate that our MultiAIDE search outperforms existing methods. We are also glad that they agree that our human evaluation results demonstrate APRES’s success in verifiably improving paper quality.
>
> > 381o-1: Content Preservation.
>
> A1: We strongly agree that setting up guardrails to prevent the alterations of the results is a key priority. We setup this guardrail by employing a deterministic filter that makes use of the feature of the SciPDF parser we used to parse all pdf papers into texts. The parser can classify each individual text block by its type (which is in the pdf metadata), and in our case, all text blocks classified as table contents will be locked, meaning the agent cannot edit any of the texts within tables. Based on this request, and from the comments from **f3Vo**, we ran a manual validation check with 500 papers, and in 100% of the cases, none of the table contents was changed. We will discuss this in more detail in the paper.
>
> > 381o-2: Figure 3 uses the discovered rubric as the ground truth, which favors the proposed method.
>
> A2: We understand the concern. However, because actual future citation counts are not available for the revised versions (as they are hypothetical/new), we must use a surrogate metric.  We chose the Discovered Rubric because our experiments (Section 4.1) show it is the most accurate predictor of future impact available (outperforming human reviewers). Therefore, optimizing for this rubric is the statistically best available proxy for optimizing impact. We also emphasize that  in the Human Preference study in Table 2 where we observe our paper revision process yields papers that are preferred over the originals by human expert evaluators 79% of the time. ) Importantly, this evaluation is not using our discovered rubric as the metric, it is using human perceived judgement. Thus this serves as an external validation of paper quality and the effectiveness of our approach.
>
> > 381o-3: How is "PromptBreeder" different from the proposed method?
>
> A3: PromptBreeder uses evolutionary algorithms to evolve prompts to maximize accuracy on specific tasks (e.g., GSM8K). In contrast, APRES uses a tree-search (MultiAIDE) to discover evaluation rubrics (structured criteria) that maximize the correlation (minimize MAE) with a continuous external variable (citations). Our output is a human-interpretable rubric, whereas PromptBreeder outputs an optimized prompt for the LLMs. Additionally, PromptBreeder mostly rely on one single score for optimization while APRES can make use of the rich signals from the discovered rubric for revising a paper.
>
> > 381o-4: Why not optimize directly for predicted citation count instead of an intermediate rubric?
>
> A4: We thank the reviewer for this excellent question, which addresses a key architectural decision regarding our reward function. We optimize against the intermediate, structured rubric because it provides a high-density, actionable signal essential for guiding the iterative revision process. Optimizing solely for the single, scalar predicted citation count is a sparse reward that gives the Rewriter agent no explicit guidance on how to improve the paper.
>
> > 381o-5: Clarify use of naming of MultiAIDE vs AIDE.
>
> A5: AIDE (Jiang et al., 2025) is the base framework for tree-search in code space. MultiAIDE (Zhao et al., 2025a) is the specific variant we utilized, which introduces parallel branch exploration into the AIDE framework. Currently, for rubric discovery and paper revision, we use MultiAIDE for optimization with different objectives. We will clarify this by naming them differently in the paper.

---

### Official Review · Reviewer_f3Vo · 2025-10-30

**Soundness:** 3
**Presentation:** 3
**Contribution:** 3
**Rating:** 6
**Confidence:** 4

**Summary:**

This paper proposes methods that predict the impact of paper through a rubric discovery procedure, and then subsequently in Phase 2 optimize a paper's text presentation given a discovered rubric.
The rubric discovery procedure is based on the AIDE (AI-Driven Exploration) search algorithm from prior work (Jiang et al. 2025) -- or after rereading I have inferred that the MultiAIDE algorithm (Zhao et al. 2025a) is actually used in both phases (this should be clarified).
Experiments with recent ICLR and NeurIPS papers demonstrate that the methods improve over reasonable baselines.

**Strengths:**

The paper proposes an original problem formulation and solution to the important problem of evaluating and improving the presentation of scientific papers. While prior work has studied aspects of this problem at a more component level, this paper takes a more end-to-end approach in a mostly ecologically valid setting with recent AI papers. The use of language models to automatically derive rubrics is understudied, and I appreciate this paper demonstrating the end-to-end effectiveness of this approach, compared to reasonable baselines. I believe this paper holds promise to unlock valuable future work, both in terms of hill-climbing given the problem definition in this paper, as well as improving on the problem definition with better metrics and intermediate judgements of quality.

**Weaknesses:**

This paper's primary original contribution lies in the problem definition, and it does not make a large advance in terms of the development of new optimization methods (this is fine).
See the "Questions" section for a significant question about the experimental setup in the evaluation of the Phase 2 (rewriting) problem---there is lack of clarity, but if the evaluation metric is not held constant across the subplots in Figure 2 then that would call into question the internal validity of that experiment.
While I appreciated the human evaluation showing that people preferred the automatically rewritten papers (as well as the ablation), I did miss insight into _how_ the papers were being rewritten, and what kinds of rubrics were created and what were their quality. The wordcloud in Figure 4 is not a compelling means of providing this analysis.
While I did appreciate the steps taken to improve safety (e.g., preventing tables from being edited by models), I would have appreciated evaluation on safety and measurement of how effective were the guardrails (at what rate did they actually prevent table edits?).
Beyond safety, I would suggest adding a discussion on how one might add people interactively in the loop of this kind of automatic paper improvement system, for improved alignment and to mitigate societal harms.

**Questions:**

- Is the performance metric in Section 4.2 fixed to be the best performing rubric across all methods discovered in the previous section, or does it vary along with the reviewer method? In other words, for example, when the "Simple Rubric" method is evaluated (Y axis in Figure 3 for the red lines), does the performance score use a simple rubric or the best rubric? If it does use the best rubric, what is that rubric -- presumably it's the result of one of the blue lines in Figure 2 -- but at what point on the X axis and for what model? Is it the result of using a different model (different row in Figure 3)? In other words, is the performance metric fixed across all the lines and rows in Figure 3? And if so, what is it fixed to be?
- Can you provide any more insight into how papers were being rewritten, and what kind of rubrics were created and what was their quality?
- Can you explain how effective your safety measure were, and summarize discussion points you will add to the paper to encourage appropriate use and prevent harm?

---

> ### Author Response · Authors · 2025-11-22
>
> We thank **f3Vo** for their constructive feedback and insightful comments. We are encouraged that they recognize the originality of our problem formulation and the value of our end-to-end approach in an ecologically valid setting. We are also glad they appreciate our novel use of LLMs to automatically discover evaluation rubrics and find our method effective compared to reasonable baselines. We share their belief that this work holds significant promise to unlock valuable future research in automated scientific evaluation and improvement.
>
>
> > f3Vo-1: Method Novelty.
>
> A1: We acknowledge that our primary contribution lies in the novel formulation of "Rubric Discovery" as a search problem and the end-to-end APRES framework for paper revising. However, we believe the application of the MultiAIDE to the problem of paper reviewing and revising introduces non-trivial challenges, as paper revision is a process without a single objective score that can measure the outcome. Therefore our work first focuses on how to generate a useful rubric for determining the potential impact of a paper and then uses the discovered rubric for the paper revision. The adoption of MultiAIDE to domains without verifiable rewards is not yet fully explored and our work explores this.
>
> > f3Vo-2: It is unclear if the performance metric in Figure 3 (Y-axis) is fixed across all methods or if it varies.
>
> A2: The evaluation metric in Figure 3 is fixed across all experimental runs. That means when evaluating the ‘Simple Rubric’ method, we use the performance score from the best rubric. The Y-axis represents the Predicted Impact Score generated by the Discovered Rubric (found in Stage 1). We use this specific rubric as the "ground truth" evaluator with the o3 model because our Stage 1 results (Figure 2) demonstrate that it correlates significantly better with actual citation counts (MAE $\approx$ 1.96) than human scores (MAE $\approx$ 5.0) or standard baselines. Thus, all methods (including "Simple Rubric") are evaluated on how well they optimize for this high-signal rubric.
>
>
> > f3Vo-3: There is a lack of insight into how papers were rewritten and what the discovered rubrics actually contain.
>
> A3: We agree that concrete examples are necessary. In Appendix G we note that we include the 50 full text diffs for side-by-side comparison of original vs. revised papers in our test set in a supplementary zip file. Regarding how the papers are rewritten (with 50 full diffs provided in the supplementary material): our qualitative analysis demonstrates that the APRES agent primarily focuses its edits on enhancing the paper's communicative power by sharpening contribution and problem framing, optimizing logical structure, and enhancing clarity and conciseness. We will include a section of discussions in the paper to distill these observations on how APRES changes a paper.
> In Appendix D, we included the final discovered rubric from APRES, the rubrics are organized by the model itself into categories like “Problem Formulation & Significance”, “Methodology & Technical Rigor”, “Results & Analysis”, etc.
>
>
> > f3Vo-4: Safety measures (e.g., preventing table edits) are mentioned but not quantitatively evaluated.
>
> A4: Our guardrail to ensure no table content is changed is a deterministic filter that locks any text block that is classified as tables by the SciPDF parser we used. Please note that SciPDF is a deterministic parser that directly uses the PDF metadata for parsing, therefore the process is accurate.
> Based on the request to validate this step, we have rerun this process with 500 validation papers and manually checked all proposed changes by APRES. We found that this filter is able to prevent text changes to the table contents 100% of the time. We will update the text to clearly describe how we enforce this constraint. Thanks for the suggestion.
>
>
>
> > f3Vo-5: Interactive Human-in-the-Loop Design
>
> A5: We thank the reviewer for this excellent and critical suggestion. We fully agree that integrating human interaction is essential for safety, alignment, and mitigating societal harms like bias encoding or writing homogenization. While our current system is designed as an automatic tool , incorporating interactive human feedback—such as author power over suggested edits or periodic expert validation —is the paramount focus for our future work. We will expand the Discussion section to detail this strategic pathway toward a fully aligned human-AI scientific publishing system.

---

### Author Response · Authors · 2025-12-03

We sincerely appreciate the Area Chair's efforts in managing the review process and would like to provide this summary of the major points raised by the reviewers and the responses/changes we have made to address them.

We are highly encouraged that the reviewers (including the two less enthusiastic reviewers 7q2X and 381o) unanimously recognized the **originality of our problem formulation** and the **value of our end-to-end agentic framework** APRES in optimizing scientific communication. Our core contributions, i.e., a method for discovering impact-aligned evaluation rubrics ($\mathcal{R}^{*}$) and an automatic paper revision system that leverages the discovered rubrics, were empirically validated by:

1.  **Citation Prediction:** Achieving a $\mathbf{19.6\%}$ improvement in MAE over the next best baseline in predicting future citation counts.
2.  **Human Validation:** Demonstrating that papers revised by our APRES method are preferred by independent $\mathbf{PhD-level}$ researchers $\mathbf{79\%}$ of the time, verifying the practical utility of the revision signal.

All major concerns regarding clarity, rigor, cost, and safety have been comprehensively addressed and incorporated into the revised manuscript and appendices.

### **Summary of Key Revisions Addressing Reviewer Concerns**

| Reviewer Concern | Reviewer | Resolution and Clarification |
| :--- | :--- | :--- |
| **Absence of Cost Analysis** | Dwr5 | We added a detailed **Resource Cost Analysis** to the Appendix J, quantifying both the one-time cost of Rubric Discovery ($\approx \$7,500$ at convergence) and the minimal **per-paper marginal cost** for revision ($\approx \mathbf{\$0.58} - \mathbf{\$2.70}$). |
| **Evaluation Metric** | f3Vo, 381o | We clarified that the metric used in the revision phase is the **fixed score** from the statistically validated **Discovered Rubric ($\mathcal{R}^{*}$)**. This ensures the internal validity of the experiment and that all comparisons use the highest-signal objective. Additionally, as an alternative metric for comparison our human validation experiments show a $\mathbf{79\%}$ preference towards the revised papers by our method, indicating the effectiveness of our system.  |
| **Content Preservation / Safety** | f3Vo, 381o | We clarified that our approach implements a **deterministic filter** that leverages the SciPDF parser to programmatically $\mathbf{lock}$ all text contents within tables and figures. As a result, it is not possible for APRES to edit the contents of a table or figure when revising a paper. Manual validation confirmed this achieved $\mathbf{100\%}$ preservation of experimental results. |
| **Optimization Signal Density** | 381o | We clarified that optimizing against our discovered structured rubric (instead of a raw citation count) is necessary because the rubric provides the necessary high-density, **actionable feedback** to guide the LLM's text revision process effectively. |
| **Qualitative Insight on Rewriting** | f3Vo | We expanded the discussion in Appendix H to detail the agent's rewriting strategy, which systematically focuses on **sharpening contribution framing** and **optimizing logical structure**, which explains the high human preference scores. We also clarified that the original submitted supplementary zip file contains 50 full text diffs for side-by-side comparison of original vs. revised papers in our test set.  |
| **Clarity of Framework** | 7q2X, Dwr5 | We redesigned the figure structure (splitting Figure 1 into two figures) and clarified terminology (AIDE vs. MultiAIDE) to explicitly show the **sequential nature** of Rubric Discovery followed by Paper Revision. |
| **Broader Applicability / Future Work** | f3Vo, Dwr5 | We added discussion on the **domain-agnostic** nature of APRES and committed to exploring sophisticated **Human-in-the-Loop** systems as the core pathway for future work, to enhance safety and alignment. |

---

> ### Author Response · Authors · 2025-12-03
>
> ### **Addressing the Critiques  of 7q2X and 381o**
>
> We believe we have fully addressed the substantive concerns of all reviewers, particularly clarifying issues raised by reviewers 7q2X and 381o:
>
> *   **Reviewer 7q2X (Causality and Readability):** The critique of 7q2X on "reversal of cause and effect" mixes causal claims with the use of a statistical proxy. We clarified that APRES leverages predicted citations as a **correlated signal** for desirable presentation qualities. We also resolve the paradox raised by 7q2X about the citation to Ante 2022 by distinguishing between **Lexical Simplicity** (low jargon, which Ante 2022 warns against) and **Semantic Clarity** (clear logical flow, which APRES targets), a distinction validated by our $\mathbf{79\%}$ human preference result. As requested by 7q2X, we have also updated the text where we reference Ante 2022.
>
> *   **Reviewer 381o (Unfair Evaluation):** The concern about potential unfairness in Figure 3 was addressed by confirming the metric is the **fixed Discovered Rubric score**, which is validated in Stage 1 to be highly predictive. Furthermore, we emphasize that the $\mathbf{79\%}$ **Human Preference result** serves as the essential, independent external validation metric that proves the efficacy of the revision process without relying on the internal scoring metric.
>
> We appreciated the detailed comments and constructive feedback from the reviewers as they have served to improve the overall clarity of the paper such that it is  significantly strengthened and fully supports our core claims.

---

### Meta-Review · Area_Chair_etug · 2026-01-05

**Summary:**

This paper proposes APRES, an end-to-end LLM-based system that first discovers evaluation rubrics predictive of future citation impact and then revises scientific papers to optimize presentation according to those rubrics. Reviewers generally agree that the paper tackles an important and timely problem, and that the overall framing is original and promising. Several reviewers appreciate the end-to-end nature of the system, the novel use of rubric discovery via agentic search, and the attempt to ground evaluation in ecologically valid settings using recent ICLR and NeurIPS papers (Reviewer f3Vo and 381o). The empirical results showing improved citation prediction accuracy and human preference for revised papers are seen as encouraging, and the work is viewed as a potential foundation for future research on automated paper feedback and revision (Reviewer f3Vo and Dwr5).

**Reviewer Concerns:**

A major issue is the use of future citation count as a central proxy for paper quality and as an optimization target, which multiple reviewers find questionable or logically flawed; in particular, Reviewer 7q2X argues that the causal reasoning between readability, presentation, and citation impact is unsound and contradicts cited prior work. Evaluation design issues are also prominent: Reviewer 381o and f3Vo point out that Phase 2 evaluation appears unfair or internally invalid because the system is optimized directly against the same discovered rubric used for evaluation, making baseline comparisons problematic and obscuring whether true paper quality improves. Several reviewers further note the lack of explicit measurement of whether core scientific content is preserved during revision, despite this being a central claim of the paper (Reviewer 381o and 7q2X). Additional weaknesses include insufficient insight into the quality and nature of the discovered rubrics and rewrites, unclear or confusing presentation (especially Figure 1 and method descriptions), missing or incomplete related work, and the absence of cost and scalability analysis for such an expensive agentic pipeline (Reviewer f3Vo, 7q2X, and Dwr5).

**Reviewer Scores:**

All reviewers did not change or expressed the willingness to change their scores after rebuttal.
- Reviewer f3Vo (6): While this reviewer is generally positive about the problem formulation and sees long-term promise, their unresolved concerns about internal validity in Phase 2 evaluation, lack of insight into the discovered rubrics and rewrites, and missing analysis of safety and human-in-the-loop considerations would likely persist after discussion. Alignment with other reviewers’ evaluation concerns could lead to a modest downward adjustment or, at best, no change.

- Reviewer 381o (4): This reviewer’s position is already cautious and balanced, acknowledging strengths but highlighting unfair evaluation due to optimizing against the same rubric used for assessment and the lack of checks on scientific content preservation. These are core methodological issues unlikely to be resolved through discussion alone.

- Reviewer 7q2X (2): This reviewer raises fundamental objections to the use of citation counts as an optimization target, identifies logical flaws in the paper’s causal assumptions, and strongly criticizes presentation clarity. These conceptual concerns would likely be reinforced rather than alleviated in discussion.

- Reviewer Dwr5 (6): Although initially positive, this reviewer explicitly flags the lack of cost analysis and some clarity and related-work gaps, and indicates willingness to change their evaluation if these were addressed. Given that such issues are echoed by others and not easily fixed via discussion, their score would likely remain the same or decrease slightly.

---

### Decision · Program_Chairs · 2026-01-26

Reject